# SLOTFORMER: UNSUPERVISED VISUAL DYNAMICS SIMULATION WITH OBJECT-CENTRIC MODELS

**Ziyi Wu**[1,2]**, Nikita Dvornik**[3,1]**, Klaus Greff**[4]**, Thomas Kipf***[4]**, Animesh Garg***[1,2]
[1] University of Toronto, [2] Vector Institute, [3] Samsung AI Centre Toronto, [4] Google Research

## ABSTRACT

Understanding dynamics from visual observations is a challenging problem that requires disentangling individual objects from the scene and learning their interactions. While recent object-centric models can successfully decompose a scene into objects, modeling their dynamics effectively still remains a challenge. We address this problem by introducing SlotFormer – a Transformer-based autoregressive model operating on learned object-centric representations. Given a video clip, our approach reasons over object features to model spatio-temporal relationships and predicts accurate future object states. In this paper, we successfully apply SlotFormer to perform video prediction on datasets with complex object interactions. Moreover, the unsupervised SlotFormer's dynamics model can be used to improve the performance on supervised downstream tasks, such as Visual Question Answering (VQA), and goal-conditioned planning. Compared to past works on dynamics modeling, our method achieves significantly better long-term synthesis of object dynamics, while retaining high quality visual generation. Besides, SlotFormer enables VQA models to reason about the future without object-level labels, even outperforming counterparts that use ground-truth annotations. Finally, we show its ability to serve as a world model for model-based planning, which is competitive with methods designed specifically for such tasks. Additional results and details are available at our Website.

## 1 INTRODUCTION

The ability to understand complex systems and interactions between its elements is a key component of intelligent systems. Learning the dynamics of a multi-object systems from visual observations entails capturing *object* instances, their appearance, position and motion, and simulating their spatio-temporal interactions. Both in robotics (Finn et al., 2016; Lee et al., 2018) and computer vision (Shi et al., 2015; Wang et al., 2017), unsupervised learning of dynamics has been a central problem due to its important practical implications. Obtaining a faithful dynamics model of the environment enables future prediction, planning and, crucially, allows to transfer the dynamics knowledge to improve downstream supervised tasks, such as visual reasoning (Chen et al., 2020b; Ding et al., 2021b), planning (Sun et al., 2022) and model-based control (Micheli et al., 2022). Yet, an effective domain-independent approach for unsupervised visual dynamics learning from video remains elusive.

One approach to visual dynamics modeling is to frame it as a prediction problem directly in the pixel space (Shi et al., 2015; Wang et al., 2017; Denton & Fergus, 2018). This paradigm builds on global frame-level representations, and uses dense feature maps of past frames to predict future features. By design, such models are object-agnostic, treating background and foreground modeling as equal. This frequently results in poorly learned object dynamics, producing unrealistic future predictions over longer horizons (Oprea et al., 2020). Another perspective to dynamics learning is through object-centric dynamics models (Kosiorek et al., 2018; van Steenkiste et al., 2018; Kossen et al., 2019). This class of methods first represents a scene as a set of object-centric features (a.k.a. slots), and then learns the interactions among the slots to model scene dynamics. It allows for more natural dynamics modeling and leads to more faithful simulation (Veerapaneni et al., 2020; Zoran et al., 2021). To achieve this goal, earlier object-centric models bake in strong scene (Jiang et al., 2019) or object (Lin et al., 2020) priors in their frameworks, while more recent methods (Kipf et al., 2020; Zoran et al., 2021) learn object interactions purely from data, with the aid of Graph Neural Networks (GNNs) (Battaglia et al., 2018) or Transformers (Vaswani et al., 2017). Yet, these approaches

independently model the per-frame object interactions and their temporal evolution, using different networks. This suggests that a simpler and more effective dynamics model is yet to be designed.

In this work, we argue that learning a system's dynamics from video effectively requires two key components: i) *strong unsupervised object-centric representations* (to capture objects in each frame) and ii) a *powerful dynamical module* (to simulate spatio-temporal interactions between the objects). To this end, we propose SlotFormer: an elegant and effective Transformer-based object-centric dynamics model, which builds upon object-centric features (Kipf et al., 2022; Singh et al., 2022), and requires no human supervision. We treat dynamics modeling as a sequential learning problem: given a sequence of input images, SlotFormer takes in the object-centric representations extracted from these frames, and predicts the object features in the future steps. By conditioning on multiple frames, our method is capable of capturing the spatio-temporal object relationships simultaneously, thus ensuring consistency of object properties and motion in the synthesized frames. We evaluate SlotFormer on four video datasets consisting of diverse object dynamics. Our method not only presents competitive results on standard video prediction metrics, but also achieves significant gains when evaluating on object-aware metrics in the long range. Crucially, we demonstrate that SlotFormer's unsupervised dynamics knowledge can be successfully transferred to downstream supervised tasks (e.g., VQA and goal-conditional planning) to improve their performance "for free". In summary, this work makes the following contributions:

1. SlotFormer: a Transformer-based model for object-centric visual simulation;

2. SlotFormer achieves state-of-the-art performance on two video prediction datasets, with significant advantage in modeling long-term dynamics;

3. SlotFormer achieves state-of-the-art results on two VQA datasets and competitive results in one planning task, when equipped with a corresponding task-specific readout module.

## 2 RELATED WORK

In this section, we provide a brief overview of related works on physical reasoning, object-centric models and Transformers, which is further expanded in Appendix A.

**Dynamics modeling and intuitive physics.** Video prediction methods treat dynamics modeling as an image translation problem (Shi et al., 2015; Wang et al., 2017; Denton & Fergus, 2018; Lee et al., 2018), and model changes in the pixel space. However, methods that model dynamics using global image-level features usually struggle with long-horizon predictions. Some approaches leverage local priors (Finn et al., 2016; Ebert et al., 2017), or extra input information (Walker et al., 2016; Villegas et al., 2017), which only help in the short term. More recent works improve modeling visual dynamics using explicit object-centric representations. Several works directly learn deep models in the abstracted state space of objects (Wu et al., 2015; Battaglia et al., 2016; Fragkiadaki et al., 2016; Chang et al., 2016). However, they require ground-truth physical properties for training, which is unrealistic for visual dynamics simulation. Instead, recent works use object features from a supervised detector as the base representation for visual simulation (Ye et al., 2019; Li et al., 2019; Qi et al., 2020; Yu et al., 2022) with a GNN-based dynamics model. In contrast to the above works, our model is completely unsupervised; SlotFormer belongs to the class of models that learn both object discovery and scene dynamics without supervision. We review this class of models below.

**Unsupervised object-centric representation learning from videos.** Our work builds upon recent efforts in decomposing raw videos into temporally aligned *slots* (Kipf et al., 2022; Kabra et al., 2021; Singh et al., 2022). Earlier works often make strong assumptions on the underlying object representations. Jiang et al. (2019) explicitly decompose the scene into foreground and background to apply fixed object size and presence priors. Lin et al. (2020) further disentangle object features to represent object positions, depth and semantic attributes separately. Some methods leverage the power of GNNs or Transformers to eliminate these domain-specific priors (Goyal et al., 2019; 2021; Veerapaneni et al., 2020; van Steenkiste et al., 2018; Creswell et al., 2021; Zoran et al., 2021). However, they still model the object interactions and temporal dynamics with separate modules; and set the context window of the recurrent dynamics module to only a single timestep. The most relevant work to ours is OCVT (Wu et al., 2021), which also applies Transformers to slots from multiple frames. However, OCVT utilizes manually disentangled object features, and needs Hungarian matching for latent alignment during training. Therefore, it still underperforms RNN-based baselines in the video prediction task. In contrast, SlotFormer is a general Transformer-based dynamics model which is ag-

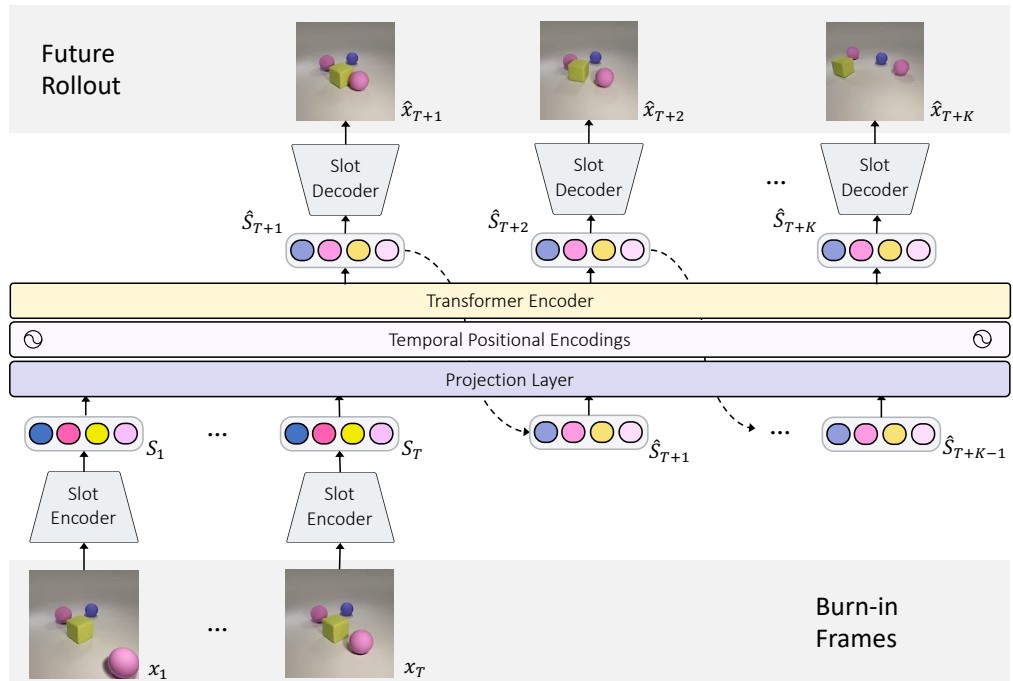

**Figure 1:** SlotFormer architecture overview. Taking multiple video frames $\{\boldsymbol{x}_t\}_{t=1}^{T}$ as input, we first extract object slots $\{\mathcal{S}_t\}_{t=1}^{T}$ using the pretrained object-centric model. Then, slots are linearly projected and added with temporal positional encoding. The resulting tokens are fed to the Transformer module to generate future slots $\{\hat{\mathcal{S}}_{T+k}\}_{k=1}^{K}$ in an autoregressive manner.

nostic to the underlying object-centric representations. It performs joint spatio-temporal reasoning over object slots simultaneously, enabling consistent long-term dynamics modeling.

**Transformers for sequential modeling.** Inspired by the success of autoregressive Transformers in language modeling (Radford et al., 2018; 2019; Brown et al., 2020), they were adapted to video generation tasks (Yan et al., 2021; Ren & Wang, 2022; Micheli et al., 2022; Nash et al., 2022). To handle the high dimensionality of images, these methods often adopt a two-stage training strategy by first mapping images to discrete tokens (Esser et al., 2021), and then learning a Transformer over tokens. However, since they operate on a regular image grid, the mapping ignores the boundary of objects and usually splits one object into multiple tokens. In this work, we learn a Transformer-based dynamics model over *slot*-based representations that capture the entire object in a single vector, thus generating more consistent future object states as will be shown in the experiments.

## 3 SLOTFORMER: OBJECT-ORIENTED DYNAMICS LEARNING

Taking $T$ video frames as inputs, SlotFormer first leverages a pre-trained object-centric model to extract object slots from each frame (Section 3.1). These slots are then forwarded to the Transformer module for joint spatio-temporal reasoning, and used to predict future slots autoregressively (Section 3.2). The whole pipeline is trained by minimizing the reconstruction loss in both feature and image space (Section 3.3). The overall model architecture is illustrated in Figure 1.

### 3.1 SLOT-BASED OBJECT-CENTRIC REPRESENTATION

We build on the Slot Attention architecture to extract slots from videos due to their strong performance in unsupervised object discovery. Given $T$ input frames $\{\boldsymbol{x}_t\}_{t=1}^{T}$, our object-centric model first extracts image features using a Convolutional Neural Network (CNN) encoder, then adds positional encodings, and flattens them into a set of vectors $\boldsymbol{h}_t \in \mathbb{R}^{M \times D_{enc}}$, where $M$ is the size of the flattened feature grid and $D_{enc}$ is the feature dimension. Then, the model initializes $N$ slots $\tilde{\mathcal{S}}_t \in \mathbb{R}^{N \times D_{slot}}$ from a set of learnable vectors ($t = 1$), and performs Slot Attention (Locatello et al., 2020) to update the slot representations as $\mathcal{S}_t = f_{SA}(\tilde{\mathcal{S}}_t, \boldsymbol{h}_t)$. Here, $f_{SA}$ binds slots to objects via iterative Scaled Dot-Product Attention (Vaswani et al., 2017), encouraging scene decomposition. To achieve temporal alignment of slots, $\tilde{\mathcal{S}}_t$ for $t \geq 2$ is initialized as $\tilde{\mathcal{S}}_t = f_{trans}(\mathcal{S}_{t-1})$, where $f_{trans}$ is the transition function implemented as a Transformer encoder.

Before training the Transformer-based dynamics model, we first pre-train the object-centric model using reconstruction loss on videos from the target dataset. This ensures the learned slots can accurately capture both foreground objects and background environment of the scene.

## 3.2 Dynamics Prediction with Autoregressive Transformer

**Overview.** Given slots $\{\mathcal{S}_t\}_{t=1}^{T}$ extracted from $T$ video frames, SlotFormer is able to synthesize a sequence of future slots $\{\mathcal{S}_{T+k}\}_{k=1}^{K}$ for any given horizon $K$. Our model operates by alternating between two steps: i) feed the slots into a Transformer that performs joint spatio-temporal reasoning and predicts slots at the next timestep, $\hat{\mathcal{S}}_{t+1}$, ii) feed the predicted slots back into the Transformer to keep generating future rollout autoregressively. See Figure 1 for the pipeline overview.

**Architecture.** To build the SlotFormer's dynamics module, $\mathcal{T}$, we adopt the standard Transformer encoder module with $N_{\mathcal{T}}$ layers. To match the inner dimensionality $D_e$ of $\mathcal{T}$, we linearly project the input sequence of slots to a latent space $G_t = \text{Linear}(\mathcal{S}_t) \in \mathbb{R}^{N \times D_e}$. To indicate the order of input slots, we add positional encoding (P.E.) to the latent embeddings. A naive solution would be to add a sinusoidal positional encoding to every slot regardless of its timestep, as done in Ding et al. (2021a). However, this would break the *permutation equivariance* among slots, which is a useful property of our model. Therefore, we only apply positional encoding at the temporal level, such that the slots at the same timestep receives the same positional encoding:

$$V = [G_1, G_2, ..., G_T] + [P_1, P_2, ..., P_T], \tag{1}$$

where $V \in \mathbb{R}^{(TN) \times D_e}$ is the resulting input to the transformer $\mathcal{T}$ and $P_t \in \mathbb{R}^{N \times D_e}$ denotes the sinusoidal positional encoding duplicated $N$ times. As we will show in the ablation study, the temporal positional encoding enables better prediction results despite having fewer parameters.

Now, we can utilize the Transformer $\mathcal{T}$ to reason about the dynamics of the scene. Denote the Transformer output features as $U = [U_1, U_2, ..., U_T] \in \mathbb{R}^{(TN) \times D_e}$, we take the last $N$ features $U_T \in \mathbb{R}^{N \times D_e}$ and feed them to a linear layer to obtain the predicted slots at timestep $T + 1$:

$$U = \mathcal{T}(V), \quad \hat{\mathcal{S}}_{T+1} = \text{Linear}(U_T). \tag{2}$$

For consequent future predictions, $\hat{\mathcal{S}}_{T+1}$ will be treated as the ground-truth slots along with $\{\mathcal{S}_t\}_{t=2}^{T}$ to predict $\hat{\mathcal{S}}_{T+2}$. In this way, the Transformer can be applied autoregressively to generate any given number, $K$, of future frames, as illustrated in Figure 1.

*Remark.* The SlotFormer's architecture allows to *preserve temporal consistency among slots* at different timesteps. To realize such consistency, we employ residual connections from $\mathcal{S}_t$ to $\hat{\mathcal{S}}_{t+1}$, which forces the Transformer $\mathcal{T}$ to apply refinement to the slots while preserving their absolute order. Owing to this order invariance, SlotFormer can be used to reason about individual object's dynamics for long-term rollout, and can be seamlessly integrated with downstream task models.

## 3.3 Model Training

Error accumulation is a key issue in long-term generation (Oprea et al., 2020). In contrast to prior works (Wu et al., 2021) that use a GPT-style causal attention mask (Radford et al., 2018), SlotFormer predicts all the slots at one timestep in parallel. Therefore, we train the model using the predicted slots as inputs, simulating the autoregressive generation process at test time. This reduces the train-test discrepancy, thus improving the quality of long-term generation as shown in Section 4.5.

For training, we use a slot reconstruction loss (in $L_2$) denoted as:

$$\mathcal{L}_S = \frac{1}{K \cdot N} \sum_{k=1}^{K} \sum_{n=1}^{N} ||\hat{s}_{T+k}^n - s_{T+k}^n||^2. \tag{3}$$

When using SAVi as the object-centric model, we also employ an image reconstruction loss to promote prediction of consistent object attributes such as colors and shapes. The predicted slots are decoded to images by the frozen SAVi decoder $f_{dec}$, and then matched to the original frames as:

$$\mathcal{L}_I = \frac{1}{K} \sum_{k=1}^{K} ||f_{dec}(\hat{\mathcal{S}}_{T+k}) - \boldsymbol{x}_{T+k}||^2. \tag{4}$$

The final objective function is a weighted combination of the two losses with a hyper-parameter $\lambda$:

$$\mathcal{L} = \mathcal{L}_S + \lambda \mathcal{L}_I. \tag{5}$$

## 4 EXPERIMENTS

SlotFormer is a generic architecture for many tasks requiring object-oriented reasoning. We evaluate the dynamics modeling capability of SlotFormer in three such tasks: video prediction, VQA and action planning. Our experiments aim to answer the following questions: **(1)** Can an autoregressive Transformer operating on slots generate future frames with both high visual quality and accurate object dynamics? (Section 4.2) **(2)** Are the future states synthesized by SlotFormer useful for reasoning in VQA? (Section 4.3) **(3)** How well can SlotFormer serve as a world model for planning actions? (Section 4.4) Finally, we perform an ablation study of SlotFormer's components in Section 4.5.

### 4.1 EXPERIMENTAL SETUP

**Datasets.** We evaluate our method's capability in video prediction on two datasets, *OBJ3D* (Lin et al., 2020) and *CLEVRER* (Yi et al., 2019), and demonstrate its ability for downstream reasoning and planning tasks on three datasets, *CLEVRER*, *Physion* (Bear et al., 2021) and *PHYRE* (Bakhtin et al., 2019). We briefly introduce each dataset below, which are further detailed in Appendix B.

*OBJ3D* consists of CLEVR-like (Johnson et al., 2017) dynamic scenes, where a sphere is launched from the front of the scene to collide with other still objects. There are 2,920 videos for training and 200 videos for testing. Following (Lin et al., 2020), we use the first 50 out of 100 frames of each video in our experiments, since most of the interactions end before 50 steps.

*CLEVRER* is similar to OBJ3D but with smaller objects and varying entry points throughout the video, making it more challenging. For video prediction evaluation, we follow Zoran et al. (2021) to subsample the video by a factor of 2, resulting in a length of 64. We also filter out video clips where there are newly entered objects during the rollout period. For VQA task, CLEVRER provides four types of questions: descriptive, explanatory, predictive and counterfactual. The predictive questions require the model to simulate future interactions of objects such as collisions. Therefore, we focus on the accuracy improvement on predictive questions by using SlotFormer's future rollout.

*Physion* is a VQA dataset containing realistic simulation of eight physical phenomena. Notably, Physion features diverse object entities and environments, making physical reasoning more difficult than previous synthetic VQA benchmarks. The goal of this dataset is to predict whether a red *agent* object will contact with a yellow *patient* object when the scene evolves. Following the official evaluation protocol, all models are first trained using unsupervised future prediction loss, then used to perform rollout on test scenes, where a linear readout model is applied to predict the answer.

*PHYRE* is a physical reasoning benchmark consisting of 2D physical puzzles. We use the BALL-tier, where the goal is to place a red ball at a certain location, such that the green ball will eventually come in contact with the blue/purple object, after the scene is unrolled in time. Following Qi et al. (2020), we treat SlotFormer as the world model and build a task success classifier on predicted object states as the scoring function. Then, we use it to rank a pre-defined 10,000 actions from Bakhtin et al. (2019), and execute them accordingly. We experiment on the within-template setting.

**Implementation Details.** We first pre-train the object-centric model on each dataset, and then extract slots for training SlotFormer. We employ SAVi (Kipf et al., 2022) on OBJ3D, CLEVRER, PHYRE, and STEVE (Singh et al., 2022) on Physion to extract object-centric features. SAVi leverages a CNN decoder to reconstruct videos from slots, while STEVE uses a Transformer-based slot decoder. STEVE can perform scene decomposition on visually complex data, but requires more memory and training time. Besides, we discovered that vanilla SAVi cannot handle some videos on CLEVRER. So we also introduce a stochastic version of SAVi to solve this problem. Please refer to Appendix C and their papers for complete details of network architectures and hyper-parameters.

### 4.2 EVALUATION ON VIDEO PREDICTION

In this subsection, we evaluate SlotFormer's ability in long-term visual dynamics simulation. We train all models on short clips cropped from videos, and rollout for longer steps during evaluation.

**Baselines.** We compare our approach with four baselines which are further described in Appendix D. We use a video prediction model *PredRNN* (Wang et al., 2017) that generates future frames based on global image features. To verify the effectiveness of slot representation, we train a VQ-VAE (Razavi et al., 2019) to tokenize images, and replace the slot in SlotFormer with patch tokens, denoted as *VQ-Former*. We also adopt the state-of-the-art generative object-centric model *G-SWM* (Lin et al., 2020),

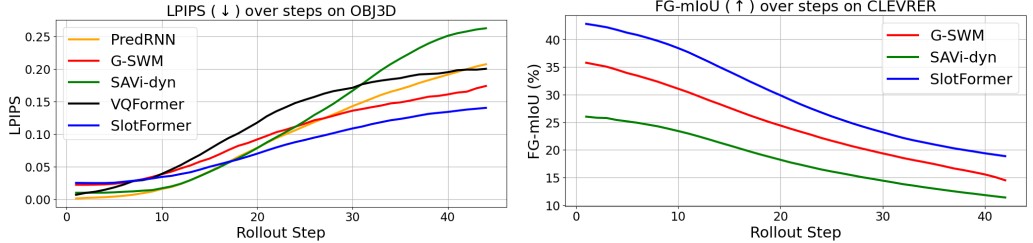

**Figure 2:** Video dynamics modeling with SlotFormer as a function of future steps. (left) Visual quality of decoded frames measured with LPIPS and (right) the quality of decoded foreground object masks with mIoU.

| Method | OBJ3D | | | CLEVRER | | |
|---|---|---|---|---|---|---|
| | PSNR ↑ | SSIM ↑ | LPIPS ↓ | PSNR ↑ | SSIM ↑ | LPIPS ↓ |
| PredRNN | **33.68** | 0.91 | 0.12 | **31.34** | **0.90** | 0.17 |
| SAVi-dyn | 32.94 | 0.91 | 0.12 | 29.77 | 0.89 | 0.19 |
| G-SWM | 31.43 | 0.89 | 0.10 | 28.42 | 0.89 | 0.16 |
| VQFormer | 30.71 | 0.86 | 0.11 | 26.80 | 0.85 | 0.18 |
| **Ours** | 32.40 | 0.91 | **0.08** | 30.21 | 0.89 | **0.11** |

**Table 1:** Evaluation of visual quality on both datasets.

| Method | AR ↑ | ARI ↑ | FG-ARI ↑ | FG-mIoU ↑ |
|---|---|---|---|---|
| SAVi-dyn | 8.94 | 8.64 | **64.32** | 18.25 |
| G-SWM | 43.98 | 57.14 | 49.61 | 24.44 |
| **Ours** | **53.14** | **63.45** | 63.00 | **29.81** |

**Table 2:** Evaluation of object dynamics on CLEVRER. All the numbers are in %.

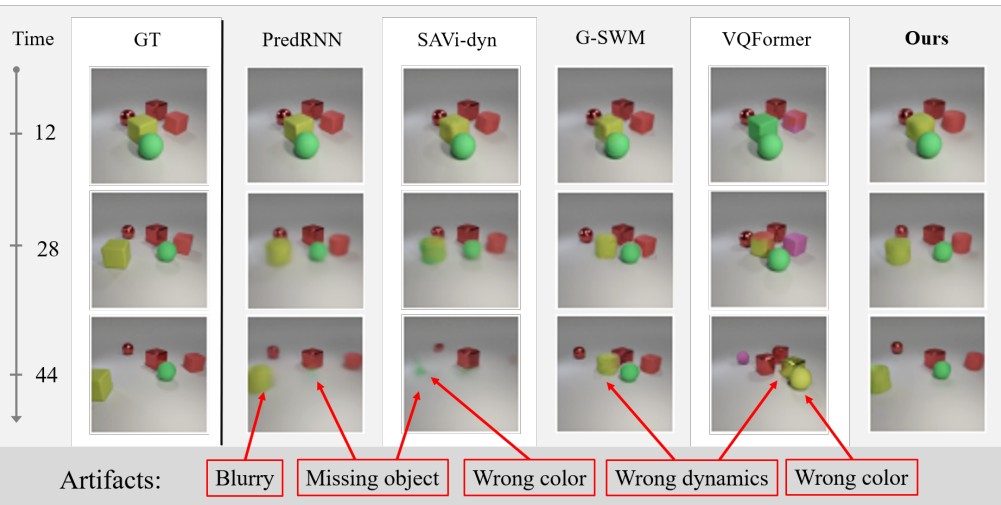

**Figure 3:** Generation results on OBJ3D. Despite higher PSNR, PredRNN and SAVi-dyn produce images with artifacts, while SlotFormer simulates sharp frames and accurate object dynamics.

which applies heavy priors. Finally, since the PARTS (Zoran et al., 2021) code is unreleased, we incorporate their Transformer-LSTM dynamics module into SAVi (denoted as *SAVi-dyn*) and train the model with the same setup. We include additional baselines (Wu et al., 2021) in Appendix E.3.

**Evaluation Metrics.** To evaluate the visual quality of the videos, we report PSNR, SSIM (Wang et al., 2004) and LPIPS (Zhang et al., 2018). As discussed in Sara et al. (2019), LPIPS captures better perceptual similarity with human than PSNR and SSIM. We focus our comparison on LPIPS, while reporting others for completeness. It is worth noting that neither of these metrics evaluate semantics in predicted frames (Yu et al., 2022). To evaluate the predicted object dynamics, we use the per-slot object masks predicted by the SAVi decoder and compare them to the ground-truth segmentation mask; same for the corresponding bounding box. We calculate the Average Recall (AR) of the predicted object boxes, and the Adjusted Rand Index (ARI), the foreground variant of ARI and mIoU termed FG-ARI and FG-mIoU of the predicted masks. We unroll the model for 44 and 42 steps on OBJ3D and CLEVRER, respectively, and report metrics averaged over timesteps.

**Results on visual quality.** Table 1 presents the results on visual quality of the generated videos. SlotFormer outperforms all baselines with a sizeable margin in terms of LPIPS, and achieves competitive results on PSNR and SSIM. We note that PSNR and SSIM are poor metrics in this setting. For example, PredRNN and SAVi-dyn score highly in these two metrics despite producing blurry objects (see Figure 3). In contrast, SlotFormer generates objects with consistent attributes throughout the rollout, which we attribute to modeling dynamics in the object-centric space, rather than in the frames directly. This is also verified in the per-step LPIPS results in Figure 2 (left). Since Slot-

| Method | per opt. (%) | per ques. (%) |
|---|---|---|
| DCL | 90.5 | 82.0 |
| VRDP | 91.7 | 83.8 |
| VRDP† | 94.5 | 89.2 |
| Aloe* | 93.1 | 87.3 |
| Aloe* + **Ours** | **96.5** | **93.3** |

**Table 3:** Predictive VQA on CLEVRER, reporting per-option (per opt.) and per-question (per ques.) accuracy. DCL and VRDP† both utilize pre-trained object detectors; * indicates our re-implementation.

| Method | Obs. (%) | Dyn. (%) | ↑ (%) |
|---|---|---|---|
| Human | 74.7 | - | - |
| RPIN* | 62.8 | 63.8 | +1.0 |
| pDEIT-lstm* | 59.2 | 60.0 | +0.8 |
| **Ours** | **65.2** | **67.1** | **+1.9** |

**Table 4:** VQA accuracy on Physion. We report the readout accuracy on observation (OBS.) and observation plus rollout (Dyn.) frames. ↑ denotes the improvement brought by the learned dynamics. Methods marked with * are our reproduced results.

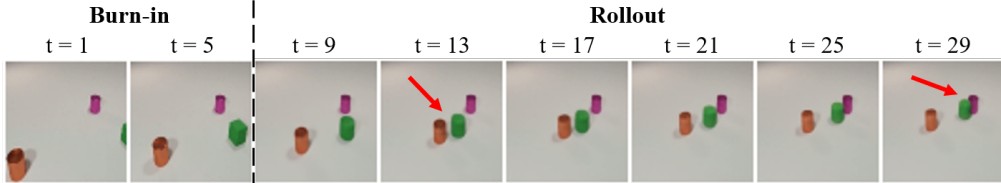

**Figure 4:** Qualitative results on CLEVRER VQA task. To answer the question "*Will the green object collide with the purple cylinder?*", SlotFormer successfully simulates the first collision between the green and the brown cylinder (t = 13), which leads to the second collision between the target objects (t = 29).

Former relies on pretrained slots, the reconstructed images at earlier steps have lower quality than baselines. Nevertheless, it achieves clear advantage at longer horizon, demonstrating superior long-term modeling ability. Although VQFormer is also able to generate sharp images, it fails to predict correct dynamics and object attributes, as also observed in previous works (Yan et al., 2021; Ren & Wang, 2022). This shows that *only a strong decoder (i.e. VQ-VAE) to generate realistic images is not sufficient* for learning dynamics. See Appendix E.1 for more qualitative results on both datasets.

**Results on object dynamics.** Here, we evaluate the quality of object bounding boxes and segmentation masks, decoded from the models' future predictions. The accuracy of the predicted object boxes and segmentation masks is summarized in Table 2 (right). Since OBJ3D lacks such annotations, and PredRNN, VQFormer cannot generate object-level outputs, we exclude it from evaluation. SlotFormer achieves the best performance on AR, ARI and FG-mIoU, and competitive results on FG-ARI. SAVi-dyn scores a high FG-ARI because its blurry predictions assign many background pixels to foreground objects, while the computation of FG-ARI ignores false positives. This is verified by its poor performance in FG-mIoU which penalizes such mistakes. We also show the per-step results in Figure 2 (right) and Appendix E.2, where our method excels at all future timesteps.

**Attention map analysis.** To study how SlotFormer leverages past information to predict the future, we visualize the self-attention maps from the Transformer $\mathcal{T}$, which is detailed in Appendix E.4.

## 4.3 VISUAL QUESTION ANSWERING

In this subsection, we show how to leverage (unsupervised) SlotFormer's future predictions to improve (supervised) predictive question answering.

**Our Implementation.** On CLEVRER, we choose *Aloe* (Ding et al., 2021a) as the base reasoning model as it can jointly process slots and texts. Aloe runs Transformers over slots from input frames and text tokens of the question, followed by an MLP to predict the answer. For predictive questions, we explicitly unroll SlotFormer and run Aloe on the predicted future slots. See Appendix C for more details. On Physion, since there is no language involved, we follow the official protocol by training a linear readout model on synthesized slots to predict whether the two objects contact. We design an improved readout model for object-centric representations, which is further detailed in Appendix C.

**Baselines.** On CLEVRER, we adopt *DCL* (Chen et al., 2020b) which utilizes pre-trained object detectors and a GNN-based dynamics model. We also choose the state-of-the-art model *VRDP* (Ding et al., 2021b), which exploits strong environmental priors to run differentiable physics engine for rollout. We report two variants of VRDP which use Slot Attention (VRDP) or pre-trained detectors (VRDP†) to detect objects. Finally, for consistency with our results, we report the performance of our re-implemented Aloe (dubbed as Aloe*).

On Physion, we select *RPIN* (Qi et al., 2020) and *pDEIT-lstm* (Touvron et al., 2021), since they are the only two methods where the rollout improves accuracy in the benchmark (Bear et al., 2021).

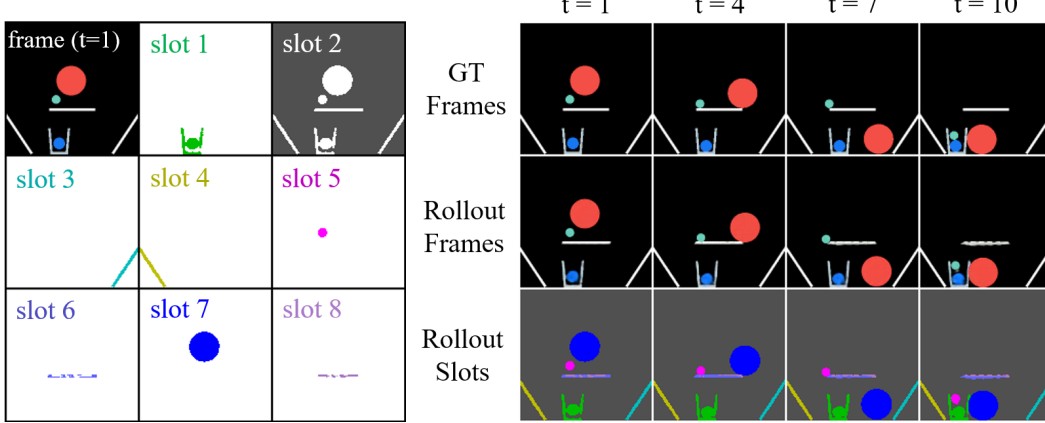

(a) Slot decomposition on the first frame    (b) Rollout results. Per-slot future predictions are color-coded.

**Figure 5:** Qualitative results on PHYRE. The goal is to place a red ball in the first frame, so that the green ball hits the blue object after rollout. We show the per-slot rollout, where SlotFormer is able to decompose the scene into individual objects, and reason their interactions to perform accurate future synthesis.

| Method | RAND | MEM | DQN | Dec [Joint] | RPIN | Dyn-DQN | SAVi | Ours |
|---|---|---|---|---|---|---|---|---|
| **Annotations** | - | - | Mask | Mask | Mask | Mask | - | - |
| **AUCCESS** | $13.7_{\pm 0.5}$ | $2.4_{\pm 0.3}$ | $77.6_{\pm 1.1}$ | $80.0_{\pm 1.2}$ | $85.2_{\pm 0.7}$ | $\mathbf{86.2}_{\pm 0.9}$ | $80.7_{\pm 1.0}$ | $82.0_{\pm 1.1}$ |

**Table 5:** AUCCESS on PHYRE-1B within-template setting. All baseline learning methods use ground-truth object segmentation masks, while SlotFormer is the only unsupervised technique learning from raw images.

RPIN is an object-centric dynamics model using ground-truth bounding boxes. pDEIT-lstm builds LSTM over ImageNet (Deng et al., 2009) pre-trained DeiT model, learning the dynamics over frame features. Since the benchmark code for Physion is not released, we reproduce it to achieve similar or better results. We also report the *Human* results from the Physion paper for reference.

**Results on CLEVRER.** Table 3 presents the accuracy on predictive questions. The dynamics predicted by SlotFormer boosts the performance of Aloe by 3.4% and 6.0% in the per option (per opt.) and per question (per ques.) setting, respectively. As a fully unsupervised dynamics model, our method even outperforms previous state-of-the-art DCL and VRDP which use supervisedly trained object detectors. On the CLEVRER public leaderboard predictive question subset, we rank first in the per option setting, and second in the per question setting. Figure 4 shows an example of our predicted dynamics, where SlotFormer accurately simulates two consecutive collision events.

**Results on Physion.** Table 4 summarizes the readout accuracy on observation (Obs.) and observation plus rollout (Dyn.) frames. SlotFormer achieves a 1.9% improvement with learned dynamics, surpassing all the baselines. See Figure 8 in the Appendix for qualitative results.

## 4.4    ACTION PLANNING

Here, we perform goal-conditioned planning inside the SlotFormer's learned dynamics model.

**Our Implementation.** Since it is possible to infer the future states from only the initial configuration on PHYRE, we set the burn-in length $T = 1$, and apply SlotFormer to generate slots $\mathcal{S}_2$ from $\mathcal{S}_1$. Then, instead of only using $\mathcal{S}_2$ to generate $\mathcal{S}_3$, we feed in both $\mathcal{S}_1$ and $\mathcal{S}_2$ for better temporal consistency. We apply this iterative overlapping modeling technique (Ren & Wang, 2022), and set the maximum conditioning length as 6. To rank the actions during testing, we train a task success classifier on future states simulated by SlotFormer, which is detailed in Appendix C. We experiment on the within-template setting, and report the *AUCCESS* metric averaged over the official 10 folds. AUCCESS measures the Area Under Curve (AUC) of the task success rate vs number of attempts curve for the first 100 attempted actions. Please refer to Appendix B for more details.

**Baselines.** We report three naive baselines from Bakhtin et al. (2019), *RAND*, *MEM* and *DQN*. We adopt *Dec [Joint]* (Girdhar et al., 2020) which employs a CNN-based future prediction model, and *RPIN* (Qi et al., 2020) as an object-centric dynamics model. Finally, Dynamics-Aware DQN (Ahmed et al., 2021) (dubbed *Dyn-DQN*) designs a task-specific loss to utilize dynamics information. Notably, all of the above methods use either ground-truth object masks or bounding boxes, while Slot-

| Method | PSNR ↑ | SSIM ↑ | LPIPS ↓ |
|---|---|---|---|
| Ours (Full Model) | **32.40** | **0.91** | **0.080** |
| Burn-in $T = 3$ | 31.26 | 0.88 | 0.093 |
| Burn-in $T = 4$ | 31.95 | 0.89 | 0.088 |
| Burn-in $T = 8$ | 32.08 | 0.90 | 0.082 |
| Trans. Layer $N_\mathcal{T} = 8$ | 32.12 | 0.89 | 0.087 |
| Naive P.E. | 32.05 | 0.90 | 0.082 |
| Teacher Forcing | 30.52 | 0.87 | 0.106 |
| No $\mathcal{L}_I$ | 31.23 | 0.88 | 0.093 |

**Table 6:** Ablation study on OBJ3D.

| Method | Improved Acc. (%) |
|---|---|
| Ours (Full Model) | **1.9** |
| Burn-in $T = 10$ | 1.0 |
| Rollout $K = 5$ | 0.5 |
| Rollout $K = 15$ | 1.9 |
| Trans. Layer $N_\mathcal{T} = 4$ | 1.3 |
| Trans. Layer $N_\mathcal{T} = 12$ | Diverge |
| Naive P.E. | 1.6 |
| Teacher Forcing | 0.2 |

**Table 7:** Ablation study on Physion.

Former learns scene dynamics without any object-level annotations. To show the performance gain by rollout, we also report *SAVi* which predicts task success purely from the initial frame's slots.

**Results on action planning.** We report the AUCCESS metric in Table 5. As an unsupervised dynamics model, SlotFormer achieves an AUCCESS score of 82.0, which improves the non-rollout counterpart SAVi by 1.3, and is on par with baselines that assume ground-truth object information as input. Figure 5 shows the entire rollout generated by our model. SlotFormer is able to capture objects with varying appearance, and simulate the dynamics of complex multi-object interactions.

## 4.5 ABLATION STUDY

In this section, we perform an ablation study to examine the importance of each component in SlotFormer on OBJ3D (Table 6) and Physion (Table 7).

**Burn-in length $T$ and rollout length $K$.** By default, we set $T = 6$, $K = 10$ for OBJ3D, and $T = 15$, $K = 10$ for Physion. On OBJ3D, the model performance first improves with more input frames, and slightly drops when $T$ further increases to 8. This might because a history length of 6 is sufficient for modeling accurate dynamics on OBJ3D. On Physion, the accuracy improves consistently as we increase $T$, until using all the observation frames. Besides, using 10 rollout frames strikes a balance between accuracy and computation efficiency. See Appendix E.5 for line plots of these results.

**Transformer (Trans.) Layer $N_\mathcal{T}$.** By default, we set $N_\mathcal{T} = 4$ on OBJ3D and $N_\mathcal{T} = 8$ on Physion. Stacking more layers harms the performance on OBJ3D due to overfitting, while improving the accuracy on Physion. This is because the dynamics on Physion is more challenging to learn. However, further increasing $N_\mathcal{T}$ to 12 makes model training unstable and the loss diverging.

**Positional encoding (P.E.).** Using a vanilla sinusoidal positional encoding which destroys the permutation equivariance among slots results in small performance drop in terms of visual quality, and a clear degradation in VQA accuracy. This is not surprising, as permutation equivariance is a useful prior for object-centric scene modeling, which should be preserved.

**Teacher forcing.** We try the teacher forcing strategy (Radford et al., 2018) by taking in ground-truth slots instead of the predicted slots autoregressively during training, which degrades the results significantly. This proves that simulating error accumulation benefits long-term dynamics modeling.

**Image reconstruction loss $\mathcal{L}_I$.** As shown in the table, the auxiliary image reconstruction loss improves the quality of the generated videos drastically. As we observe empirically, $\mathcal{L}_I$ helps preserve object attributes (e.g., color, shape), but has little effect on object dynamics. Thus, we do not apply $\mathcal{L}_I$ on Physion, due to the large memory consumption of STEVE's slot decoder.

## 5 CONCLUSION

In this paper, we propose SlotFormer, a Transformer-based autoregressive model that enables consistent long-term dynamics modeling with object-centric representations. SlotFormer learns complex spatio-temporal interactions between the objects and generates future predictions of high visual quality. Moreover, SlotFormer can transfer unsupervised dynamics knowledge to downstream (supervised) reasoning tasks which leads to state-of-the-art or comparable results on VQA and goal-conditioned planning. Finally, we believe that unsupervised object-centric dynamics models hold great potential for simulating complex datasets, advancing world models, and reasoning about the future with minimal supervision; and that SlotFormer is a new step towards this goal. We discuss the limitations and potential future directions of this work in Appendix F.

## REPRODUCIBILITY STATEMENT

All of our methods are implemented in PyTorch (Paszke et al., 2019), and can be trained on servers with 4 modern GPUs in less than 5 days, enabling both industrial and academic researchers. To ensure the reproducibility of our work, we provide detailed descriptions of how we process each dataset in Appendix B, the implementation details and hyper-parameters of the models we use in Appendix C, and sources of the baselines we compare with in Appendix D. To facilitate future research, we will release the code of our work and the pre-trained model weights alongside the camera ready version of this paper.

### ACKNOWLEDGMENTS

We would like to thank Wei Yu, Tianyu Hua for general advice and feedback on the paper, Xiaoshi Wu, Weize Chen for discussion of Transformer model implementation and training, and Jiaqi Xi, Ritviks Singh, Qinxi Yu, Calvin Yu, Liquan Wang for valuable discussions and support in computing resources.

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

## A  ADDITIONAL RELATED WORK

**Physical reasoning for dynamics modeling.** Instead of explicitly encoding physical laws to deep models and estimating latent variables from inputs (Wu et al., 2015; 2016), recent approaches implicitly infer them by modeling object interactions in the scene (Chang et al., 2016; Battaglia et al., 2018; 2016; Sanchez-Gonzalez et al., 2018; Li et al., 2019). VIN (Watters et al., 2017) employs a CNN to encode video frames into a multi-channel 1D tensor, where each channel represents an object. It enforces a fixed mapping between object identity and feature channel, which cannot generalize to different number of objects with varying appearance. CVP (Ye et al., 2019) leverage object bounding boxes to crop input image and apply CNNs to extract object-centric representations. Since object features are extracted from raw image patches, it ignores the context information and thus cannot model the interactions between objects and the environment. RPIN (Qi et al., 2020) instead uses RoIPooling (Girshick, 2015) to extract object features from image feature maps, which contains background information. However, these methods rely on ground-truth object-level annotations for training, which are often unavailable. In contrast, SlotFormer pre-trains unsupervised object-centric models on unlabeled videos. It ensures accurate decomposition of foreground objects and background environment, laying the foundation for building powerful dynamics models.

**Dynamics modeling in object-centric representation learning.** R-NEM (van Steenkiste et al., 2018) is the first end-to-end object-centric model to reason about objects and their interactions from pixel observations alone. It extracts object features from raw observations and uses an interaction function in the form of a GNN to model interactions. SCALOR (Jiang et al., 2019) scales the SQAIR (Kosiorek et al., 2018) model to work on scenes with multiple moving objects. It introduces a background module to model the image background separately. It also equips each object with a depth property to handle occlusions. STOVE (Kossen et al., 2019) incorporates a GNN-based dynamics model into SuPAIR (Stelzner et al., 2019) to reason object interactions, where object representations are explicitly disentangled into positions, velocities and appearance. Similarly, OP3 (Veerapaneni et al., 2020) learns pairwise relationship between objects based on a symmetric assumption. G-SWM (Lin et al., 2020) combines the key properties of the above methods and proposes a unified framework for accurate dynamics prediction. A hierarchical latent modeling technique is utilized to handle the multi-modality of the scene dynamics. Leveraging the power of Transformers, OAT (Creswell et al., 2021) directly learns to align slots extracted from each frame to gain temporal consistency and perform slot interactions. However, the temporal dynamics is still modeled by an LSTM (Hochreiter & Schmidhuber, 1997) module. Similarly, PARTS (Zoran et al., 2021) employs the same Transformer-LSTM module from OAT. It utilizes the Slot-Attention (Locatello et al., 2020) mechanism to detect objects and relies on a fixed independent prior to achieve stable future rollout performance. OCVT (Wu et al., 2021) is the most relevant work to SlotFormer. It also applies Transformer over slots from multiple frames and performs future prediction in an autoregressive manner. However, OCVT still disentangles its underlying object features into position, depth and semantic information. It also relies on a Hungarian matching algorithm to achieve temporal alignment of slots. As a result, OCVT is inferior to G-SWM in terms of future rollout. Compared to previous works, SlotFormer is a general Transformer-based dynamics model that is agnostic to the object-centric representations it builds upon. It does not assume any explicit disentanglement of the object property, while still can handle the object interactions well. Without the use of RNNs or GNNs, we achieve state-of-the-art dynamics modeling ability.

**Transformers.** With the prevalence of Transformers in the NLP field (Vaswani et al., 2017; Kenton & Toutanova, 2019), there have been tremendous efforts in introducing it to computer vision tasks (Dosovitskiy et al., 2020; Carion et al., 2020; Liu et al., 2021). Our method is highly motivated by previous works in Transformer-based autoregressive image and video generation (Esser et al., 2021; Chen et al., 2020a; Yan et al., 2021; Nash et al., 2022; Ren & Wang, 2022; Wu et al., 2022) and their applications in reasoning (Ding et al., 2021a; Mondal et al., 2023). VQGAN (Esser et al., 2021) first pretrains a VQ-VAE (Razavi et al., 2019) that can map images to discrete tokens and tokens back to images. Then, a GPT-like Transformer model is trained to autoregressively predict the input tokens for image generation. Transframer (Nash et al., 2022) instead discretizes video frames using Discrete Cosine Transform (DCT), and learns an autoregressive Transformer over these sparse representations from multiple frames. The design of SlotFormer is mostly related to (Ren & Wang, 2022), which also uses image tokens from multiple frames to enable consistent long-term view synthesis. Different from these works, our mapping step maps images to object-centric representations, preserving the identity of objects and is independent of the input image resolution.

## B  DATASET DETAILS

**OBJ3D (Lin et al., 2020).** The videos in this dataset are generated by first placing 3 to 5 static objects in the scene, and then launching a sphere from the front of the scene to collide with those objects. Compared to CLEVRER, the objects in OBJ3D occupy more pixels in images, have less collisions and occlusions, and are all visible in the scene at the beginning of the videos.

**CLEVRER (Yi et al., 2019).** The videos in this dataset contain static or moving objects at the beginning, and there will be various new objects entering the scene from random directions throughout the video. The smaller size and more diverse interactions of objects make CLEVRER more challenging than OBJ3D. We obtain the ground-truth segmentation masks from their official website, which are used to generate object bounding boxes. We calculate the Average Recall (AR) with an IoU threshold of $50\%$ for the predicted object boxes and the Adjusted Rand Index (ARI) for the object masks. We also report a variant of ARI and mIoU which only focus on foreground objects termed FG-ARI and FG-mIoU as done in the SAVi paper (Kipf et al., 2022).

As a Visual Question Answering (VQA) dataset, CLEVRER consists of four types of questions generated by template-based programs, namely, descriptive, explanatory, predictive and counterfactual. The latter three types of questions are multiple-choice questions, where the VQA model needs to classify whether each choice is correct.

**Physion (Bear et al., 2021).** This dataset contains eight physical scenarios, each falls under a common physical phenomenon, such as rigid- and soft-body collisions, falling, rolling and sliding motions. The foreground objects used in the simulation vary in categories, textures, colors and sizes. It also uses diverse background as the scene environment, and randomize the camera pose in rendering the videos. Overall, this dataset presents more complex visual appearance and object dynamics compared to other synthetic VQA datasets. Therefore, we apply the recently proposed powerful object-centric model, STEVE (Singh et al., 2022), to extract slots on this dataset.

Physion splits the videos into three sets, namely, *Training*, *Readout Fitting* and *Testing*. We truncate all videos by 150 frames as most of the interactions end before that, and sub-sample the videos by a factor of 3 for training the dynamics model. Following the official evaluation protocol, the dynamics models are first trained on videos from the Training set under future prediction loss. Then, they observe the first 45 frames of videos in the Readout Fitting and Testing set, and perform rollout to generate future scene representations (e.g. feature maps for image-based dynamics models, or object slots for SlotFormer). A linear readout model is trained on observed and rollout scene representations from the Readout Fitting set to classify whether the two cued objects (one in red and one in yellow) contact. Finally, the classification accuracy of the trained readout model on the Testing set scene representations is reported. Please refer to their paper (Bear et al., 2021) for detailed description of the evaluation.

**PHYRE (Bakhtin et al., 2019).** We study the PHYRE-B tier in this paper, which consists of 25 templates of tasks. Tasks within the same template share similar initial configuration of the objects. There are two evaluation settings, namely, *within-template*, where training and testing tasks come from the same templates, and *cross-template*, where train-test tasks are from different templates. We simulate the videos in 1 FPS as done in previous works (Bakhtin et al., 2019; Qi et al., 2020).

In our preliminary experiment, we discovered that SAVi usually fails to detect objects in light color (e.g. the green and gray balls). We hypothesize that this is because the $L_2$ norm of light colors is close to white (the color of background), so pixels in light colors receive gradients of small magnitude during optimization, leading to worse segmentation results. Since improving object-centric models is not the focus of this paper, we choose a simple workaround by changing the background color from white to black.

To solve a task at test time, models need to determine the size of a red ball and the location to put it in the scene, such that the green ball touches the blue/purple object for more than 3 seconds after the scene evolves. Following (Qi et al., 2020), we train models to score a pre-defined 10,000 actions when applied to the current task (by rendering the red ball in the scene), and execute them according to the ranking. The evaluation metric, *AUCCESS*, is a weighted sum of the Area Under Curve (AUC) of the success percentage vs number of attempts curve for the first 100 attempts. See their paper (Bakhtin et al., 2019) for detailed explanation of the metrics.

| Dataset | | OBJ3D | CLEVRER | Physion | PHYRE |
|---|---|---|---|---|---|
| | Base Model | SAVi | SAVi | STEVE | SAVi |
| | Image Resolution | $64 \times 64$ | $64 \times 64$ | $128 \times 128$ | $128 \times 128$ |
| Slot Model | Number of Slots $N$ | 6 | 7 | 6 | 8 |
| | Slot Size $D_{slot}$ | 128 | 128 | 192 | 128 |
| | Batch Size | 64 | 64 | 48 | 64 |
| | Training Steps | 80k | 200k | 460k | 370k |
| | Burn-in Steps $T$ | 6 | 6 | 15 | 1 |
| | Rollout Steps $K$ | 10 | 10 | 10 | 10 |
| | Latent Size $D_e$ | 128 | 256 | 256 | 256 |
| Transformer | Number of Layers $N_{\mathcal{T}}$ | 4 | 4 | 8 | 8 |
| | Loss Weight $\lambda$ | 1 | 1 | 0 | 0 |
| | Batch Size | 128 | 128 | 128 | 64 |
| | Training Steps | 200k | 500k | 250k | 300k |

**Table 8:** Variations in model architectures and training settings on different datasets.

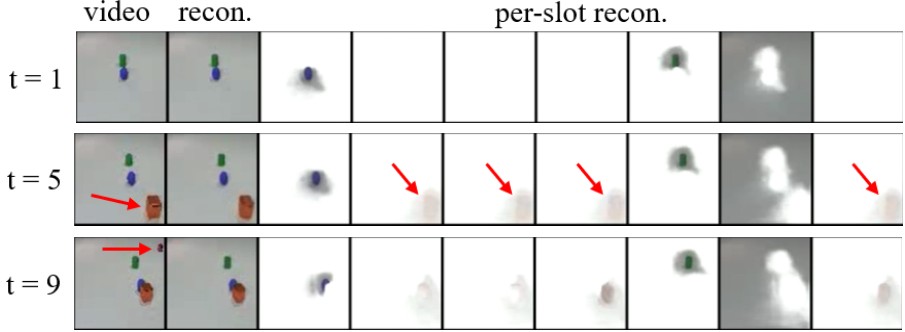

**Figure 6:** Illustration for missing objects of vanilla SAVi on CLEVRER videos. There are two objects at the beginning of this video (top). When the red cube enters the scene, all 4 empty slots attend to this object, resulting in object sharing (middle). When another object enters the scene from the top right corner, SAVi does not have empty slots to detect it (bottom). As a result, this object is ignored by the model.

## C  IMPLEMENTATION DETAILS

We provide more implementation details of our method in this section. Table 8 lists the hyper-parameters used in our experiments to facilitate reproduction.

**SAVi.** We reproduce the unconditional version of SAVi in PyTorch (Paszke et al., 2019) to perform unsupervised object discovery. Specifically, we use the same CNN encoder, decoder, Slot Attention based corrector and Transformer based predictor as their experiments on CATER, except on PHYRE the spatial broadcast size of the decoder is $16 \times 16$ to better capture small objects. The slot size is 128 and the training video clip length is 6 on all the datasets. We adopt Adam (Kingma & Ba, 2015) as the training optimizer. We use the same warmup and decay learning rate schedule which first linearly increases from 0 to $2 \times 10^{-4}$ for the first $2.5\%$ of the total training steps, and then decrease to 0 in a cosine annealing strategy. We perform gradient clipping with a maximum norm of $0.05$.

**Stochastic SAVi.** As stated in the main paper, vanilla SAVi sometimes fails to capture newly entered objects in a video, and we detail the reason and our solution as follows. We use 7 slots for SAVi on CLEVRER which has a maximum of 6 objects in the scene. Imagine a video with 4 objects $\{O_i\}_{i=1}^4$ at the beginning. Let us assume SAVi captures the objects in the first 4 slots and the background in the 5th slot. This leads to two empty slots $s_6$ and $s_7$, which are very similar with L2 distance $||s_6 - s_7||^2$ generally smaller than $0.05$. Consequently, when there is a new object $O_5$ enters the scene, $s_6$ and $s_7$ will both attend to it, resulting in object sharing between slots. Now, if there is another object $O_6$ entering the scene, there will be no "free" slot to detect this new object. Therefore, $O_6$ will be ignored by SAVi, until one of the previous object leaves the scene. An example is shown in Figure 6. This issue occurs only on CLEVRER because all the objects are presented in videos from the beginning in other datasets. Besides, SAVi did not experiment on datasets with multiple newly entered objects [1], and thus they did not observe such problem.

---

[1] confirmed with the authors of SAVi

From our analysis, the issue stems from the similarity of empty slots, which is because of the permutation equivariance of slots. To break the symmetry, we introduce stochasticity to slots initialized from previous timestep. Specifically, we modify the slot transition function by applying a two-layer MLP with Layer Normalization (Ba et al., 2016) to predict the mean and log variance of $\tilde{\mathcal{S}}_{t+1}$:

$$(\mu_{t+1}, \log \sigma_{t+1}^2) = \text{MLP}(f_{trans}(\mathcal{S}_t)). \tag{6}$$

Then, we sample from this distribution to get $\tilde{\mathcal{S}}_{t+1} \sim \mathcal{N}(\mu_{t+1}, \log \sigma_{t+1}^2)$ for performing Slot Attention with visual features at frame $t + 1$.

To enforce this stochasticity, we apply a KL divergence loss on the predicted distribution. Since we do not regularize the mean of $\tilde{\mathcal{S}}_{t+1}$, the loss only penalizes the log variance with a prior value $\hat{\sigma}$:

$$\begin{aligned} \mathcal{L}_{KL}^{t+1} &= D_{\text{KL}}(\mathcal{N}(\mu_{t+1}, \log \sigma_{t+1}^2) \,||\, \mathcal{N}(\mu_{t+1}, \log \hat{\sigma}^2)) \\ &= \log \frac{\hat{\sigma}}{\sigma_{t+1}} + \frac{\sigma_{t+1}^2}{2 \cdot \hat{\sigma}^2} - \frac{1}{2}, \end{aligned} \tag{7}$$

which will be averaged over all input timesteps. We set $\hat{\sigma} = 0.1$ which produces enough randomness to break the symmetry without destroying the temporal alignment of slots. With this simple modification, we can detect all the objects throughout the video. We use the same strategy as SAVi to train the stochastic SAVi model on CLEVRER under a combination of the frame reconstruction loss and the KL divergence loss, where the later one is weighted by a factor of $1 \times 10^{-4}$.

**STEVE.** We reproduce the $128 \times 128$ input resolution version of STEVE. Different from the paper, we adopt a two-stage training strategy by first pre-training a discrete VAE (Singh et al., 2021) to convert images into patches tokens, and then train the slot model to reconstruction these tokens. We found this strategy lead to more stable training in our experiments. Other training settings are the same as their original implementation.

**Transformer.** We follow BERT (Kenton & Toutanova, 2019) to implement our model by stacking multiple transformer encoder blocks. The number of self-attention head is 8 and the hidden size of FFN is $4 \times D_e$. We adopt the Pre-LN Transformer (Xiong et al., 2020) design as we empirically find it easier to optimize. We train our model using the Adam optimizer. The initial learning rate is $2 \times 10^{-4}$ and decayed to 0 in a cosine schedule. We also adopt a linear learning rate warmup strategy during the first $5\%$ of training steps. We do not apply gradient clipping or weight decay during training. On OBJ3D and CLEVRER, we apply both the slot reconstruction loss $\mathcal{L}_S$ and image reconstruction loss $\mathcal{L}_I$ for training. On Physion, we do not apply $\mathcal{L}_I$ due to the large memory consumption of STEVE's Transformer-based decoder. Similarly, we do not apply $\mathcal{L}_I$ on PHYRE since the image resolution is $128 \times 128$ and the spatial broadcast size of the SAVi decoder is set as $16 \times 16$, which consumes lots of GPU memory.

**VQA model on CLEVRER.** To jointly process object slots and question texts, we employ Aloe (Ding et al., 2021a) as the base VQA model given its strong performance on CLEVRER. Given a video and a question, Aloe first leverages pre-trained object-centric model to extract slots from each frame, and a text tokenizer to convert questions to language tokens. Then, it concatenates slots and text tokens, and forward them to a reasoning module, which is a stack of $N_{Aloe}$ Transformer encoder, to perform joint reasoning and predict the answer.

We re-implement Aloe in PyTorch. Following their training settings, we reproduce the results with a smaller Transformer reasoning module, $N_{Aloe} = 12$, while the original implementation uses $N_{Aloe} = 28$. This is because we use SAVi which produces higher quality and temporally consistent slots compared to the MO-Net (Burgess et al., 2019) they used. When integrating with SlotFormer, we unroll our dynamics model to predict slots at future timesteps, and feed them to Aloe to answer predictive questions. For other types of questions, the process is the same as the original Aloe.

**Readout model on Physion.** Permutation equivariance is an important property of object-centric models, which is also preserved by SlotFormer. Simply concatenating slots and forwarding it through a fully-connected (FC) layer degrades the performance, since the prediction changes according to input slot orders. To build a compatible readout model, we leverage the max-pooling operation which is invariant to the input permutations (Qi et al., 2017). Besides, to better utilize the object-centric representation, we draw inspiration from Relation Networks (Santoro et al., 2017) to explicitly reason over pairs of objects. Specifically, we concatenate every two slots and apply FC on it, then the outputs are max-pooled over all pairs of slots and time to obtain the final prediction.

In our experiments, we discovered that training readout models on the entire rollout videos (150 frames) leads to severe overfitting. This is because of the error accumulation issue in long-term video prediction, where the model overfits to artifacts introduced at later timesteps. Therefore, we only fit the readout network to the first 75 frames of the video. We evaluate baselines with the same readout model and training setting for fair comparison, which also improves their performance.

**Task success classifier on PHYRE.** We train a task success classifier to score an action for the current task. Specifically, we concatenate predicted slots with a learnable `CLS` token, add temporal positional encoding, and process them using a Transformer encoder. Then, we apply a two-layer MLP to output the score from the features corresponding to the `CLS` token. Such design also ensures the predicted score is invariant to the order of input slots.

## D  BASELINES

We detail our implementation of baselines in this section.

**PredRNN (Wang et al., 2017)** is a famous video prediction model leveraging spatio-temporal LSTM to model scene dynamics via global frame-level features. We adopt the online official implementation [2]. The models are trained until convergence for 16 epochs and 6 epochs on OBJ3D and CLEVRER, respectively. We adopt the same training settings as their original paper.

**VQFormer.** To show the effectiveness of object-centric representations, we design a baseline that replace the object slots with Vector Quantized (VQ) patch tokens. We first pre-train a VQ-VAE (Razavi et al., 2019) on video frames to convert patches to discrete tokens. We adopt the implementation of VQ-VAE from VQ-GAN (Esser et al., 2021) [3], where we set the number of tokens per-frame as $4 \times 4 = 16$, and the codebook size as 4096. The autoregressive Transformer follows the design in SlotFormer. We also tried the GPT-like training strategy (i.e. causal masking) as done in Micheli et al. (2022), but did not observe improved performance.

**G-SWM (Lin et al., 2020)** unifies several priors in previous object-centric models and is shown to achieve good results on various simple video datasets. It constructs a background module to process the scene context, disentangles object features to positional and semantic information, explicitly models occlusion and interaction using depth and GNN module, and performs hierarchical latent modeling to deal with the multi-modality over time. We use the online official implementation [4]. We train the model for 1M steps on both datasets, and select the best weight via the loss on the validation set. Our re-trained model achieves slightly better results than their pretrained weight on the OBJ3D dataset. Therefore, we also adopt the this training setting on CLEVRER.

**SAVi-dyn.** Since neither the code of PARTS (Zoran et al., 2021) nor its testing dataset (PLAY-ROOM) is released, and PARTS did not try future prediction task on CLEVRER, we try our best to re-implement it to compare with SlotFormer under our settings. Inspired by its design, we replace the Transformer predictor in SAVi (Kipf et al., 2022) with the Transformer-LSTM dynamics module in PARTS. The model is trained to observe initial burn-in frames, and then predict the slots as well as the reconstructed image of the rollout frames using the dynamics module. We use a learning rate of $1 \times 10^{-4}$ and train the model for 500k steps. The other training strategies follow SAVi.

We do not compare with OCVT (Wu et al., 2021) because it underperforms G-SWM even on simple 2D datasets, while SlotFormer outperforms G-SWM under all the settings.

**Aloe (Ding et al., 2021a)** runs Transformers over object slots and text tokens of the question to perform reasoning. The official code [5] was written in TensorFlow (Abadi et al., 2016), so we re-implement it in PyTorch to fit in our codebase. We adopt the same model architecture and hyper-parameters as the original paper, except that we use 12 layer Transformer encoder while they use 28, as our SAVi slot representations are more powerful than their MO-Net (Burgess et al., 2019) slots. We train the model for 250k steps.

**RPIN (Qi et al., 2020)** is an object-centric dynamics model using ground-truth object bounding boxes of the burn-in frames. For a fair comparison on Physion dataset with SlotFormer, we also

---

[2] https://github.com/thuml/predrnn-pytorch

[3] https://github.com/CompVis/taming-transformers

[4] https://github.com/zhixuan-lin/G-SWM

[5] https://github.com/deepmind/deepmind-research/tree/master/object_attention_for_reasoning

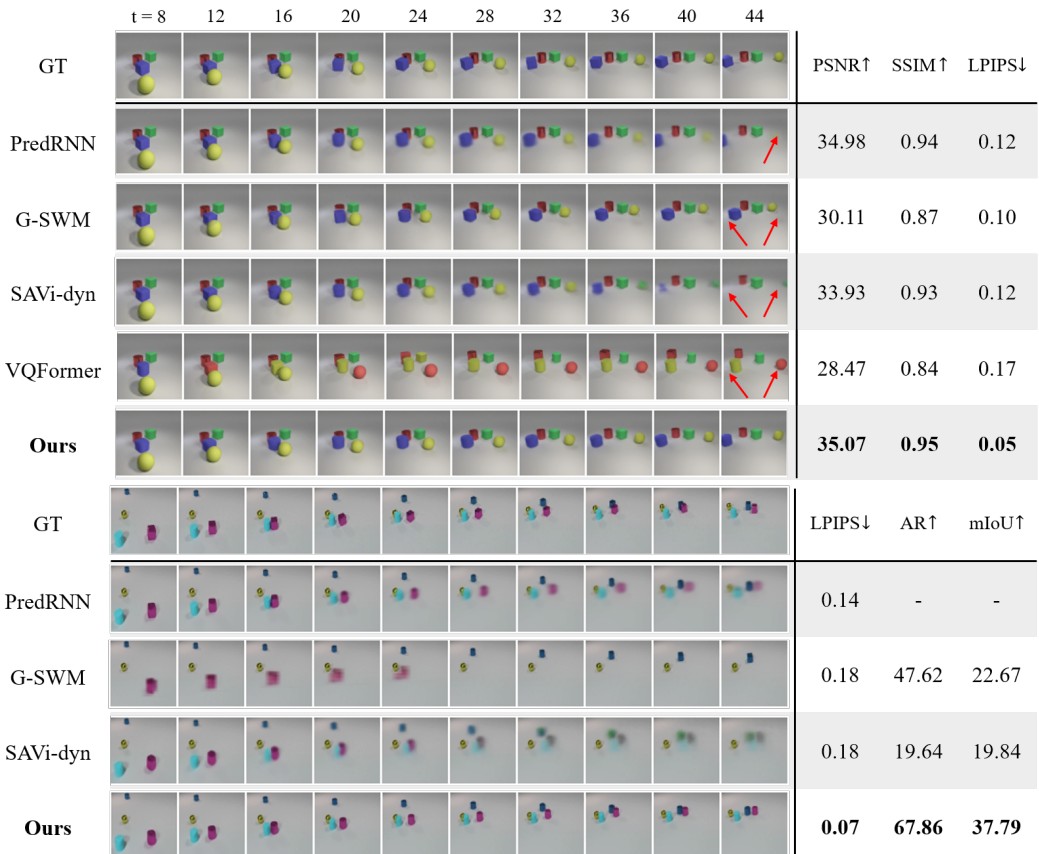

**Figure 7:** Generation results on OBJ3D (top) and CLEVRER (bottom). On the right, we report metrics measuring the visual quality and object trajectory of the visualized rollouts for each model.

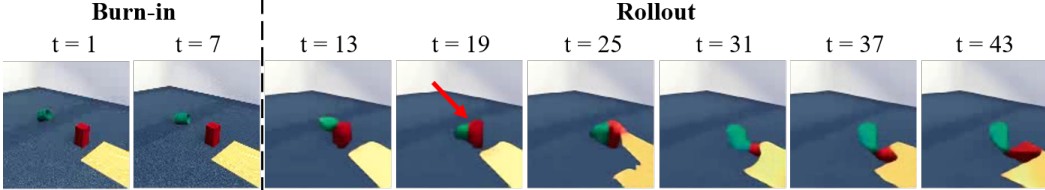

**Figure 8:** Qualitative results on Physion VQA task. To answer the question "*Will the red object contact with the yellow object?*", SlotFormer successfully simulates the falling of the red box. The low visual quality of the predicted frames is due to STEVE's Transformer-based decoder, which is not designed for pixel-space reconstruction. Nevertheless, they still preserve the correct motion of objects.

apply our improved readout model (see Appendix C) on RPIN. We adopt the online official implementation [6] and train it on Physion dataset for 300k steps. As shown in Table 4, our reproduced readout accuracy is much higher than the reported result in the benchmark.

**pDEIT-lstm** applies an LSTM over frame features extracted by ImageNet (Deng et al., 2009) pre-trained DEIT (Touvron et al., 2021) model. We follow the original implementation and use the DEIT model provided by timm (Wightman, 2019) (`deit_base_patch16_224`). We frozen the DEIT model and train the LSTM for 100k steps.

For other baselines, we simply copy the numbers from previous papers.

# E MORE EXPERIMENTAL RESULTS

## E.1 QUALITATIVE RESULTS

**Video prediction.** Figure 7 (top) shows additional qualitative results on OBJ3D. SlotFormer achieves excellent generation of the object trajectories thus very low LPIPS score. However, its

---

[6]https://github.com/HaozhiQi/RPIN

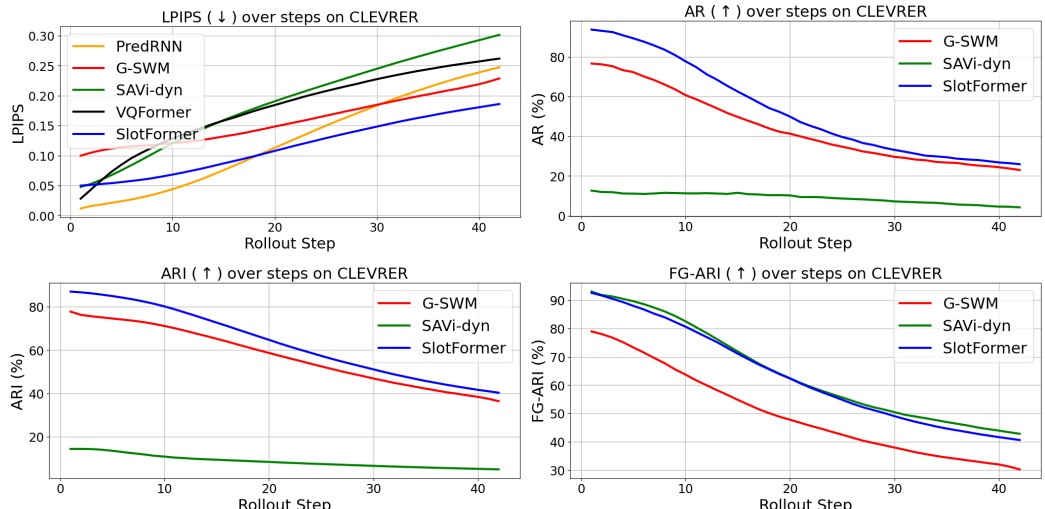

**Figure 9:** Comparison of the object dynamics of the generated videos at each rollout step on CLEVRER. We report FG-API (left) and FG-mIoU (right) of the segmentation masks.

| Method | Predictive | |
|---|---|---|
| | per opt. (%) | per ques. (%) |
| DCL | 90.5 | 82.0 |
| VRDP | 91.7 | 83.8 |
| VRDP† | 94.5 | 89.2 |
| Aloe | 93.5 | 87.5 |
| Aloe* | 93.1 | 87.3 |
| Aloe* + **Ours** | **96.5** | **93.3** |

**Table 9:** Predictive VQA on CLEVRER, reporting per-option (per opt.) and per-question (per ques.) accuracy. DCL and VRDP† both utilize pre-trained object detectors; * indicates our re-implementation.

| Method | Obs. (%) | Dyn. (%) | ↑ (%) |
|---|---|---|---|
| Human | 74.7 | - | - |
| RPIN | 54.3 | 55.7 | + 1.4 |
| RPIN* | 62.8 | 63.8 | + 1.0 |
| pDEIT-lstm | 59.9 | 60.5 | + 0.6 |
| pDEIT-lstm* | 59.2 | 60.0 | + 0.8 |
| **Ours** | **65.2** | **67.1** | **+ 1.9** |

**Table 10:** VQA accuracy on Physion. We report the readout accuracy on observation (OBS.) and observation plus rollout (Dyn.) frames. ↑ denotes the improvement brought by the learned dynamics. Methods marked with * are our reproduced results.

PSNR and SSIM are still close to PredRNN and SAVi-dyn, which blurs the moving objects into the background in later frames. This again proves that LPIPS are superior metrics for measuring the generated videos. Besides, G-SWM can also preserve the object identity because it leverages complex priors such as depth to model occlusions. Nevertheless, its simulated dynamics are still worse than our Transformer model. Finally, VQFormer is able to generate sharp images without blurry objects because of its strong VQ-VAE decoder. However, the object properties such as colors are not consistent, and the simulated dynamics are erroneous.

We present a visual result on CLEVRER in Figure 7 (bottom). The objects are smaller in size and have longer term dynamics, making it much harder than OBJ3D. PredRNN and SAVi-dyn still generate blurry objects at later steps. G-SWM sometimes cannot detect objects newly entering the scene because of the limited capacity of its discovery module. In contrast, SlotFormer builds Transformer on SAVi slots, enabling both accurate object detection and precise dynamics modeling. This is also verified by the object-aware metrics AR and mIoU we show in the figures.

See Figure 13 and Figure 14 for more qualitative results in the video prediction task.

**VQA.** Figure 8 shows a qualitative result on Physion dataset, where SlotFormer successfully synthesizes the contact of the red object and the yellow ground. Note the low quality of the predicted frames is due to the STEVE's Transformer-based decoder, which is not designed for pixel-space reconstruction [7]. Improving the decoder design is beyond the scope of this paper.

---

[7]In our experiments, even the reconstruction results of STEVE on Physion videos are of low quality.

| Method | Descriptive | Explanatory | | Predictive | | Counterfactual | | Average |
|---|---|---|---|---|---|---|---|---|
| | | per opt. | per ques. | per opt. | per ques. | per opt. | per ques. | |
| Aloe* + **Ours** | 95.17 | 98.04 | 94.79 | 96.50 | 93.29 | 90.63 | 73.78 | 89.26 |

**Table 11:** Accuracy on different questions and average results on CLEVRER. Numbers are in %.

| Method | Collide | Contain | Dominoes | Drape | Drop | Link | Roll | Support | Average |
|---|---|---|---|---|---|---|---|---|---|
| SlotFormer | 69.3 | 63.3 | 55.6 | 66.7 | 62.7 | 69.3 | 70.7 | 77.3 | 67.1 |

**Table 12:** Accuracy breakdown for all eight scenarios on Physion. Numbers are in %.

| Method | Fold ID | | | | | | | | | | Average |
|---|---|---|---|---|---|---|---|---|---|---|---|
| | 0 | 1 | 2 | 3 | 4 | 5 | 6 | 7 | 8 | 9 | |
| SlotFormer | 83.1 | 83.2 | 81.0 | 81.2 | 81.2 | 83.0 | 82.6 | 80.0 | 83.0 | 81.8 | $82.0_{\pm 1.1}$ |

**Table 13:** AUCCESS for all 10 folds on PHYRE.

| Method | OBJ3D | | | CLEVRER | | |
|---|---|---|---|---|---|---|
| | PSNR ↑ | SSIM ↑ | LPIPS ↓ | PSNR ↑ | SSIM ↑ | LPIPS ↓ |
| OCVT | 31.08 | 0.88 | 0.13 | 27.96 | 0.87 | 0.18 |
| Slot-LSTM | 32.15 | 0.90 | 0.09 | 29.79 | 0.88 | 0.13 |
| **Ours** | **32.40** | **0.91** | **0.08** | **30.21** | **0.89** | **0.11** |

**Table 14:** Evaluation of visual quality on both datasets.

| Method | AR ↑ | ARI ↑ | FG-ARI ↑ | FG-mIoU ↑ |
|---|---|---|---|---|
| OCVT | 36.19 | 51.23 | 40.87 | 20.57 |
| Slot-LSTM | 48.52 | 59.58 | 58.42 | 27.84 |
| **Ours** | **53.14** | **63.45** | **63.00** | **29.81** |

**Table 15:** Evaluation of object dynamics on CLEVRER. All the numbers are in %.

### E.2 QUANTITATIVE RESULTS

**Video prediction.** We show the per-step FG-ARI and FG-mIoU results in Figure 9. The sophisticated priors in G-SWM prevents it from scaling to scenes with multiple objects and complex dynamics. Since SAVi-dyn generates blurry objects, it produces many false positives in the segmentation masks. Instead, SlotFormer preserves the object identity and achieves high scores in both metrics over long rollout steps.

**VQA.** Table 9 and Table 10 present the complete results of the original and our reproduced performance of baselines, as well as ours on both VQA datasets. We report the performance of our Aloe with SlotFormer model on all four question types of CLEVRER in Table 11. We report the per-scenario accuracy of SlotFormer on Physion rollout setting in Table 12.

**Planning.** Table 13 shows the AUCCESS of SlotFormer for all 10 folds on PHYRE.

### E.3 COMPARISON WITH ADDITIONAL BASELINES

In this section, we compare SlotFormer with two additional baselines, namely *OCVT* (Wu et al., 2021) and *Slot-LSTM*, in the video prediction task on OBJ3D and CLEVRER datasets.

**OCVT** builds Transformers over SPACE (Lin et al., 2019) which applies heavy priors in their framework. In addition, it requires a Hungarian alignment step of slots for loss computation. Therefore, OCVT underperforms G-SWM (Lin et al., 2020) in long-term generation. Since its code is not released, we reproduce it based on SPACE [8], and adopt its setting in the video prediction task.

As shown in Table 14 and Table 15, OCVT underperforms SlotFormer in both visual quality of videos and accuracy of object dynamics. This is because object slots from SAVi is more powerful than SPACE, and SlotFormer naturally enjoys the temporal alignment of slots.

**Slot-LSTM** trains a Transformer-LSTM dynamics module from PARTS (Zoran et al., 2021) over the same pre-trained object slots as SlotFormer. We adopt the same Transformer module as SlotFormer, but only feed in slots at a single timestep, thus only modeling the spatial interaction of objects. The Transformer is followed by a per-slot LSTM to learn the temporal dynamics.

As shown in Table 14 and Table 15, SlotFormer outperforms Slot-LSTM in all the metrics, especially on the more challenging CLEVRER dataset. This indicates the importance of joint spatial-temporal reasoning over a larger context window. Despite having the same Transformer module, Slot-LSTM

---

[8]https://github.com/zhixuan-lin/SPACE

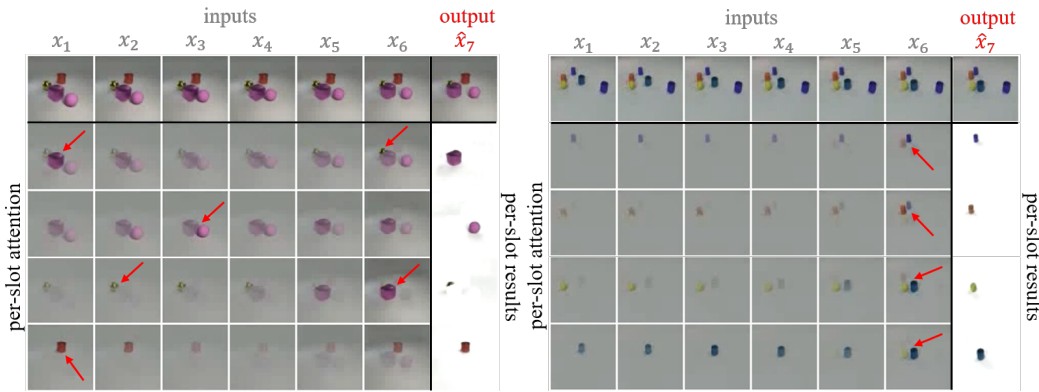

**Figure 10:** Example attention map visualization on OBJ3D (left) and CLEVRER (right). Our model takes in slots from $\{x_i\}_{i=1}^6$ (column 1-6) to predict slots of $\hat{x}_7$ (column 7). We show images at the first row and the per-slot future reconstructions at the rightmost column. The body of the table shows the per-slot attention of SlotFormer when predicting $\hat{\mathcal{S}}_7$, with the arrows pointing at the regions of high importance for predicting the future slot in the same row. Zoom in for better viewing.

limits the context window of its recurrent module to only a single timestep. Therefore, it still generates videos with blurry objects and inconsistent dynamics over the long horizon.

### E.4 ATTENTION ANALYSIS

In this section, we analyze the visual cues in the input frames that SlotFormer utilizes to make future predictions. We do so by visualizing the attention map from the last self-attention layer in the transformer $\mathcal{T}$. More precisely, given the last $T$ encoded frames $\{\mathcal{S}_t\}_{t=1}^T$, we are predicting the future slots $\hat{\mathcal{S}}_{T+1}$. Denote the attention scores from $\hat{s}_{T+1}^i$ to $s_t^j$ as $a_{t,j}^i$, where $i, j \in [1, N]$ and $N$ is the number of slots. At each timestep $t$ and for each future slot $i$, we obtain spatial attention maps $o_t^i$ over input frames $x_i$ as a weighted combination of the slot reconstructions as follows:

$$o_t^i = \sum_{j=1}^N a_{t,j}^i \cdot (m_t^j \odot y_t^j), \tag{8}$$

which indicates the regions of $x_t$ SlotFormer attends upon when predicting $\hat{s}_{T+1}^i$.

Figure 10 (left) presents one example from OBJ3D, where the purple cube just collided with the purple sphere, and is about to hit the yellow sphere. When predicting the purple cube, the model focuses on the past collision event in $\{x_i\}_{i=1}^4$, and highlights the yellow sphere in $x_6$. For the purple sphere, the Transformer only looks at the purple cube because it will not hit the yellow sphere. Since the yellow sphere becomes heavily occluded in $x_6$, SlotFormer attends to earlier frames, while predicting its future motion based on the purple cube. This indicates that SlotFormer can handle occlusions or disappearing of objects during burn-in frames by attending to other timesteps where the objects are visible, and using that information to infer the properties and motion of objects. Finally, the red cylinder merely looks at itself because it is not involved in the collisions.

Figure 10 (right) illustrates one example from CLEVRER. We only analyze the left side of the images since there is no object interaction in the right part. There are two collision events (the purple cylinder hitting the orange cylinder, and the yellow sphere hitting the blue cube) happening in $x_7$, and SlotFormer successfully captures their interactions in the attention maps. In general, we found the attention maps in CLEVRER less clear than those in OBJ3D, due to the smaller object size. Nevertheless, the Transformer can still detect correct cues to reason their future motion.

### E.5 ABLATION STUDY

Figure 11 shows the effect of burn-in and rollout length on SlotFormer's performance as line plot for better clarity. On OBJ3D, the LPIPS first improves as we use more burn-in frames, and then degrades after reaching a peak at $T = 6$. We do not ablate the rollout length as it is fixed according to the evaluation setting. On Physion, the accuracy gain increases consistently with more burn-in frames as it provides more context information. Therefore, we choose the maximum length $T = 15$ according to the number of observed frames available at test time. Finally, the accuracy grows as we use more rollout frames during training, and plateaus after $K = 10$.

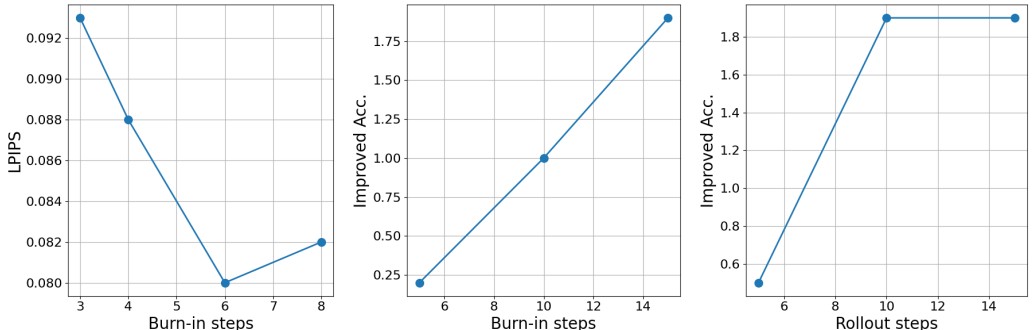

**Figure 11:** Ablation study on burn-in and rollout length of SlotFormer. We show the LPIPS of generated videos on OBJ3D (left), and the improved VQA accuracy by rollout on Physion (middle, right).

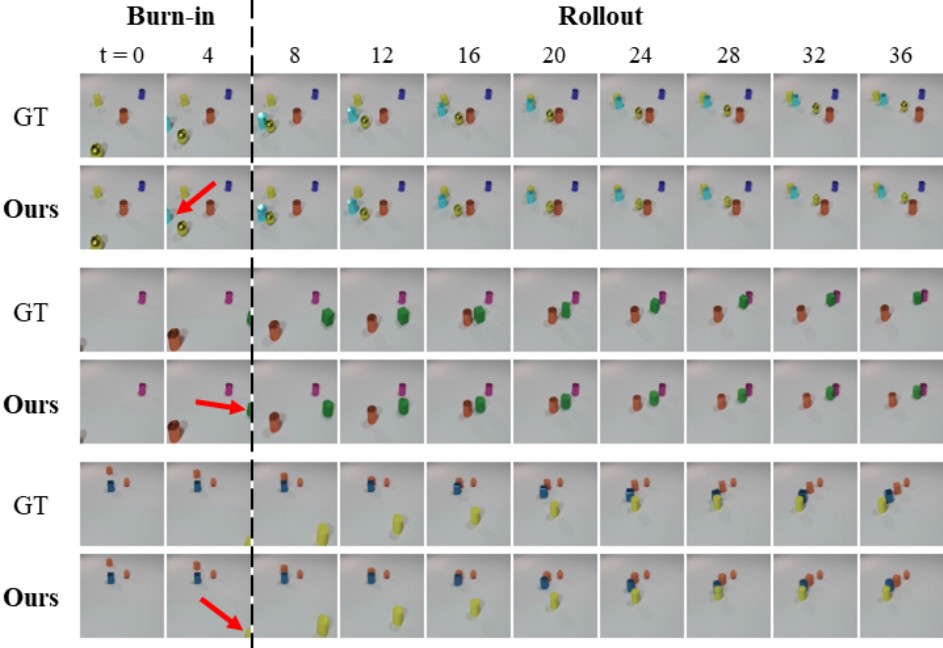

**Figure 12:** Videos generated by SlotFormer on CLEVRER where some objects are not visible at the initial frame, but enters the scene during burn-in frames (marked by red arrows). SAVi is able to detect these new objects, and SlotFormer can still simulate accurate future dynamics for these objects.

### E.6 NEW OBJECTS DURING BURN-IN FRAMES

One concern regarding SlotFormer is that, if some objects are not visible at the first timestep, but enters the scene during burn-in frames, can our model still be able to simulate their dynamics? Figure 12 shows a few examples with new objects appearing during burn-in steps on the CLEVRER dataset. Since the object-centric model SAVi is able to detect these new objects, SlotFormer can still reason their dynamics based on frames where they are visible, and simulate accurate future states. This is an important property of SlotFormer as disappearing and (re-)appearing of objects are common in real world videos.

## F LIMITATIONS AND FUTURE WORKS

**Limitations.** SlotFormer currently builds upon pretrained object-centric models. This family of methods still fail to scale up to real world data [9], preventing our application to real world videos as well. Besides, the two-stage training strategy harms the model performance at the early rollout

---

[9]STEVE (Singh et al., 2022) can work on real world videos such as traffics, where ground usually shares the same color, and looks distinct from the vehicles. Also, STEVE's Transformer-based slot decoder cannot generate images of high visual quality on complex datasets.

steps as shown in Figure 2. It is interesting to explore joint training of the base object-centric model and the Transformer dynamics module, which could potentially benefit the performance of both models. Finally, current SlotFormer model works in a deterministic manner, and thus cannot model the uncertainty of future dynamics, which is common in real world videos.

**Future Works.** We only experiment on unconditional future prediction in this paper. In the future, we plan to extend SlotFormer to conditional generation tasks, such as action-conditioned generation as done in Zoran et al. (2021). Recent works have shown success in this direction by converting conditional inputs to tokens and feeding them to the Transformer (Ren & Wang, 2022; Tevet et al., 2022). Another direction is to simplify the training process by learning scene decomposition and temporal dynamics jointly. This may allow the object-centric model to leverage long-term motion cues for unsupervised object discovery (Yang et al., 2021). Finally, it is important to enable Slot-Former to learn the multi-modality of future dynamics for stochastic video prediction. This is key to modeling real world videos faithfully (Oprea et al., 2020).

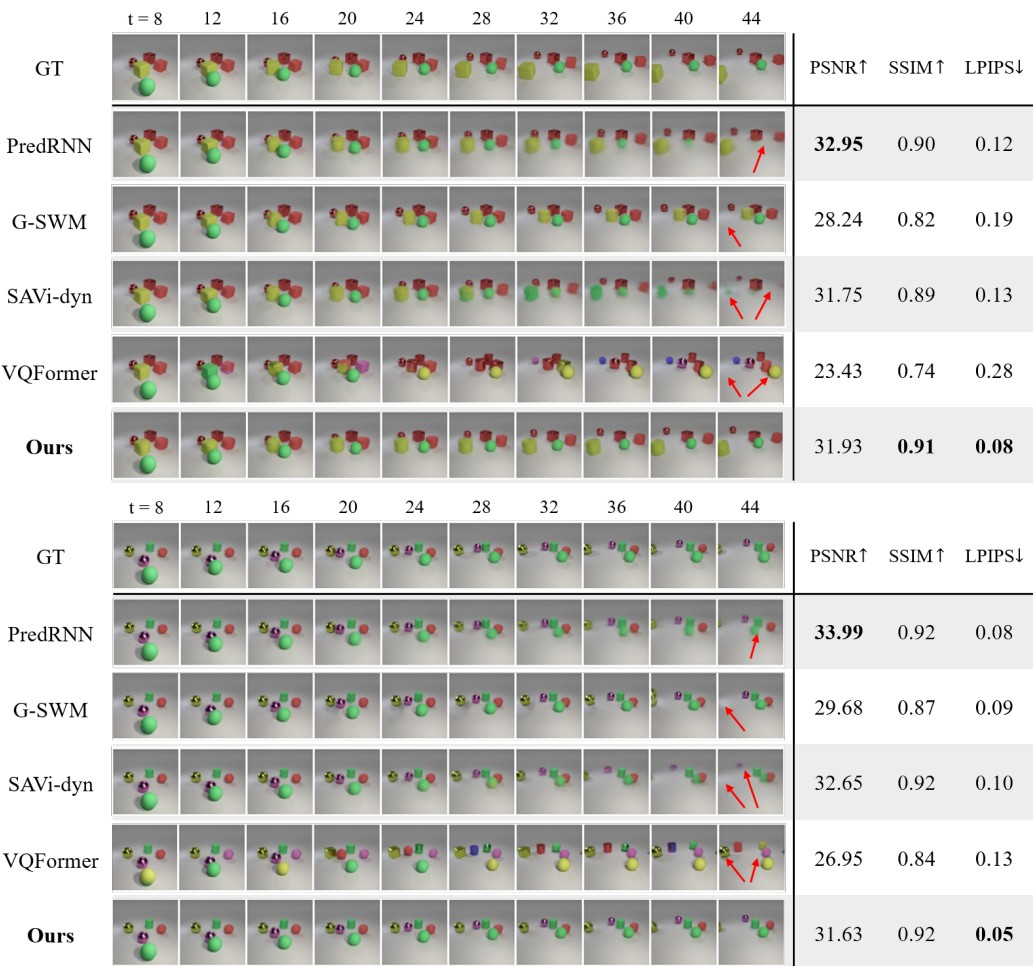

**Figure 13:** More qualitative results on OBJ3D.

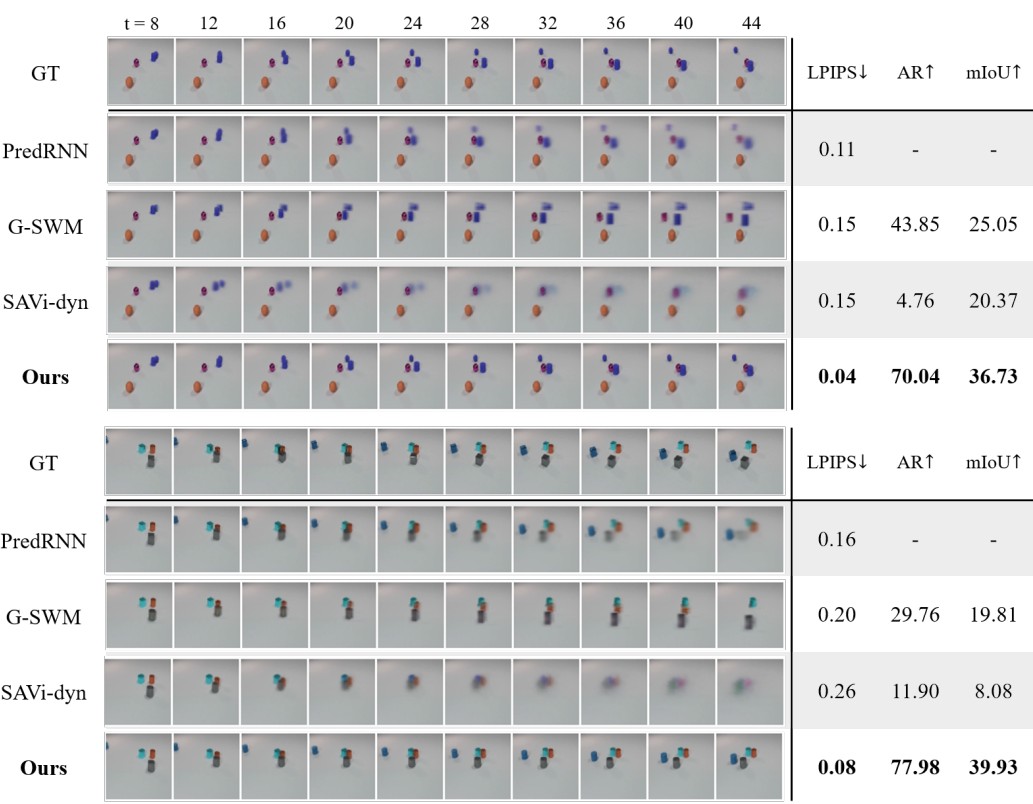

**Figure 14:** More qualitative results on CLEVRER.

