# OpenReview forum: "SlotFormer: Unsupervised Visual Dynamics Simulation with Object-Centric Models"
_ICLR.cc/2023/Conference — ICLR 2023 poster_

### Official Review · Reviewer_HGtg · 2022-10-20

**Confidence:** 4
**Correctness:** 3
**Technical Novelty And Significance:** 2
**Empirical Novelty And Significance:** 3
**Recommendation:** 6

**Clarity, Quality, Novelty And Reproducibility:**

The paper is clear and the supplementary material provides detailed implementation details, which I believe reduces the difficulty of reproducibility. The paper is not novel in terms of the model design whereas it is novel in the configuration of the experiments.

**Strength And Weaknesses:**

Strength:
+ The idea is neat and simple. It fully takes the advantage of the previous object-centric model (i.e., SAVI).
+ The experiment results and visualization strongly show the effectiveness of the method. For example, the emerging property of long-term modeling is well supported by the plot in Figures 2 and 4.
+ The paper writing is easy to follow.

Weakness:
- The novelty of the architecture is limited. The solely new module in the paper on top of the SAVI is the Transformer blocks. Also, I am confused about why this proposed module excels in long-term modeling. The intuition here is not strong. Does the temporal-dependent position embedding work? Or, does the transformer architecture itself help the long-term modeling? If that is the case, what is the result without naive positional embedding (when the RNN-based model is adopted)? Maybe the authors would be better to highlight it more in the paper.
- I am confused why would "error accumulation issue" benefit the model. Is there any related work to analyze the phenomenon (the error propagation is crucial in the sequential/autoregressive model)? I don't think it can be treated as common sense for the readers and requires a more reasonable explanation.

**Summary Of The Paper:**

The paper introduces a core module to model the interaction of frames and predict the consequent states in an object-centric way. Mainly built on the SAVI [1], the proposed module operates on the objects of burn-in frames and generates future rollout autoregressively. In specific, it adopts layers of transformer block added with positional embedding for retaining permutation equivariance to model the object-dynamic representations. Thorough experiments demonstrate that the approach achieves competitive performance on not only video prediction tasks but also some reasoning tasks (action planning and visual question answering).

[1] Conditional Object-Centric Learning from Video. ICLR 2022.

**Summary Of The Review:**

Overall, the paper shows a promising application for slot attention in object-dynamic modeling. However, some intuition the paper tries to convey is not straightforward. For example, the long-term property. Which design of the model makes it excel in long-term prediction? Also, the technical novelty is limited in terms of the architecture. Thus, my initial recommendation is borderline reject.

---

> ### Author Response · Authors · 2022-11-11
> **Response to Reviewer HGtg**
>
> Thank you for your review and insightful comments.
>
>
> **Q1: The novelty of SlotFormer architecture.**
>
> A: We thank the reviewer for sharing their opinion. In fact, we view the simplicity of SlotFormer as an advantage rather than a shortcoming. SlotFormer is easy to implement, reproduce and at the same time it achieves state-of-the-art results on a number of diverse downstream tasks, such as future prediction, VQA and planning. We believe that for exactly that reason SlotFormer is likely to be welcomed by the community and make a real impact. We would also like to note that both reviewer eCuX and reviewer yrhY recognize SlotFormer as elegant and effective, which can serve as a starting point for future methods in more advanced settings.
>
> Besides, we also made several technical contributions in the paper:
> - Instead of using the naive positional encoding, we design a temporal positional encoding, which can preserve the permutation-equivariance property of object-centric models. This eliminates the use of the Hungarian algorithm for object alignment in previous work [1]. As shown in the ablation study (see Section 4.5), our temporal positional encoding improves the model performance in both video prediction and downstream VQA tasks;
> - We design a stochastic version of SAVi to handle videos where some objects are not visible in the first frame. In Appendix C, we analyze why vanilla SAVi fails in this case, and describe our improvement in detail. In our preliminary experiments, naively combining SAVi and a Transformer does not work on the CLEVRER dataset.
>
> [1] Wu, Yi-Fu, Jaesik Yoon, and Sungjin Ahn. "Generative Video Transformer: Can Objects be the Words?." ICML. 2021.
>
>
> **Q2: The intuition why SlotFormer outperforms baselines in long-term modeling & Implementing a RNN-based baseline to better convey the intuition.**
>
> A: The Transformer module with multi-step inputs is the key to SlotFormer’s success in long-term dynamics modeling. RNN-based models such as G-SWM [1] and PARTS [2] set the context window of its dynamics module to only a single timestep. As discussed in previous works [3, 4], this harms the consistency of long-term prediction. In contrast, SlotFormer leverages a Transformer to reason over slots from multiple frames. The larger context window leads to better long-term consistency.
>
> This can be verified by the ablation study where we vary the size of the context window (i.e. the number of burn-in steps). In Appendix E.5 Figure 11 of the revised paper ([this figure](https://i.imgur.com/WmiXBEA.png)), you can see that with the increase of burn-in steps, the model performance improves (LPIPS goes down and Accuracy goes up) until it reaches a peak. This proves that a reasonably large context window is key to accurate long-term dynamics modeling.
>
> As suggested by the reviewer, we implement a baseline called Slot-LSTM, which also builds upon the same pre-trained object slots, but uses the Transformer-LSTM dynamics module from PARTS. It first uses a Transformer to model **single-step** object interactions, and then uses an LSTM to model temporal dynamics. We test this baseline in video prediction on OBJ3D and CLEVRER datasets. As shown in Appendix E.3 of the revised paper, Slot-LSTM underperforms SlotFormer in all the metrics. This indicates that a Transformer that performs multi-step reasoning is the key to the success of our method in long-term modeling.
>
>
> [1] Lin, Zhixuan, et al. "Improving generative imagination in object-centric world models." ICML. 2020.
>
> [2] Zoran, Daniel, et al. "PARTS: Unsupervised segmentation with slots, attention and independence maximization." ICCV. 2021.
>
> [3] Wu, Yi-Fu, Jaesik Yoon, and Sungjin Ahn. "Generative Video Transformer: Can Objects be the Words?." ICML. 2021.
>
> [4] Oprea, Sergiu, et al. "A review on deep learning techniques for video prediction." T-PAMI. 2020.

---

> > ### Author Response · Authors · 2022-11-11
> > **Response to Reviewer HGtg (Part 2)**
> >
> > **Q3: Clarification of the “error accumulation issue”.**
> >
> > A: We apologize for the confusion caused by the text. We have clarified this in the paper now.  As shown in several papers [1, 2, 3], error accumulation is a key issue in long-term video prediction. This is because during training, the model predicts future frames based on ground-truth video frames, while at test time, it has to autoregressively predict the future by taking in frames generated by itself. Such discrepancy degrades model performance at long horizons.
> >
> > To improve long-term prediction, we propose to simulate the error accumulation issue during training. Instead of using all ground-truth slots as model inputs (i.e. teacher forcing), SlotFormer also takes in slots generated by itself to predict further future slots. Similar technique was also used in previous works [4, 5]. It reduces the domain gap of model inputs between training and testing, and thus significantly improves the performance as shown in the ablation study (Section 4.5). We have clarified this point more clearly in Section 3.3 of the revised paper.
> >
> > [1] Wang, Yunbo, et al. "Predrnn: Recurrent neural networks for predictive learning using spatiotemporal lstms." NeurIPS. 2017.
> >
> > [2] Babaeizadeh, Mohammad, et al. "Stochastic variational video prediction." ICLR. 2018.
> >
> > [3] Oprea, Sergiu, et al. "A review on deep learning techniques for video prediction." T-PAMI. 2020.
> >
> > [4] Bengio, Samy, et al. "Scheduled sampling for sequence prediction with recurrent neural networks." NeurIPS. 2015.
> >
> > [5] Ren, Xuanchi, and Xiaolong Wang. "Look Outside the Room: Synthesizing A Consistent Long-Term 3D Scene Video from A Single Image." CVPR. 2022.

---

> > > ### Comment · Reviewer_HGtg · 2022-11-17
> > > **Response to Rebuttal**
> > >
> > > Thanks for the efforts the authors put in during the rebuttal. I think my concern has been addressed mostly. Thus, I would like to raise my score to 6 and recommend acceptance. Good luck!

---

> > > > ### Author Response · Authors · 2022-11-17
> > > > **Re: Response to Rebuttal**
> > > >
> > > > Thank you for updating your review! We are happy to hear that we were able to address most of your concerns.

---

### Official Review · Reviewer_yrhY · 2022-10-24

**Confidence:** 4
**Correctness:** 4
**Technical Novelty And Significance:** 4
**Empirical Novelty And Significance:** 4
**Recommendation:** 8

**Clarity, Quality, Novelty And Reproducibility:**

- To my knowledge, the work is novel. The closest work is OCVT. However, Slotformer has important advantages over OCVT. OCVT builds on the SPACE family of object representations while SlotFormer builds on the slot-attention family of object representations — the former requiring heavy priors while the latter is more general and has also been shown to work well on visually complex scenes e.g. in SAVi/STEVE. Furthermore, OCVT seems to require explicit Hungarian alignment to compute the prediction loss while Slotformer naturally enjoys aligned slots from SAVi/STEVE.
- The paper is clear and high-quality.
- Authors also promise to release the code.

**Strength And Weaknesses:**

### Pros

1. The model is novel as it makes use of slot-based representations of SAVi/STEVE and models complex long-term spatiotemporal dependencies using a transformer for predicting the next-step slots.
2. Shows good future predictions. Shows benefits on downstream tasks such as VQA and action planning. Also shows useful ablations.
3. The model is simple and effective and thus has the potential to become the go-to model for further exploration in this line.
4. Has a nice property that the future slot predictions preserve the original order of the slots.
5. An interesting observation is that even though the predicted frames of STEVE have low visual quality, the slot representations were still useful to learn future prediction at the latent level and do well on the VQA downstream task.
6. VQ-former is said to provide poor dynamics suggesting that object-centric representation can be key for good learning of dynamics.
7. Writing is quite clear.

### Weaknesses/Questions

1. I understand that the positional encodings are duplicated over all $N$ slots for a given time-step $t$. However, I am wondering how the positional encodings $P_t$ are set for a given $t$? Are these sinusoidal embeddings in terms of $t$? If so, then would the model generalize in predicting much longer time horizons than those seen in training? Also, are the episodes randomly cropped to make fixed-length clips before training, or are full episodes used directly during training?
2. Currently, the fact that VQFormer is not performing well is only mentioned in the text and the evaluation metrics (SSIM/PSNR) were said to be not ideal by the authors. So, it would be good to show some videos of VQFormer not performing well in dynamics prediction relative to SlotFormer. This would better highlight that object-centric slots are indeed important. Also, showing some generated videos of SlotFormer for all the datasets can be helpful to see the visual quality of the generations.
3. How beneficial is it to perform explicit roll-outs for the downstream tasks? For instance, in VQA, what would be the performance if the downstream model receives only the burn-in frames without doing the explicit future roll-out? It may be good to show a comparison. A similar question may be asked about the PHYRE action-planning task i.e. what if the slots from burn-in frames are directly used to predict the task-success score?

### Minor Comments/Questions

1. In the ablation, the effect of burn-in and the roll-out length may be shown as a line plot.
2. Have authors tested action-conditioned generation in SlotFormer and how was it implemented?

**Summary Of The Paper:**

The paper proposes a transformer-based dynamics model that can be trained on top of pre-trained slot representation of a video. The model first pre-trains SAVi/STEVE to obtain slots from video frames. It then trains an auto-regressive transformer to learn to predict future slots given the past slots.

**Summary Of The Review:**

I think the paper can be accepted in its current form. However, clarifying/answering some of the above questions would be good and I would be happy to raise the score.

---

> ### Author Response · Authors · 2022-11-11
> **Response to Reviewer yrhY**
>
> We thank the reviewer for the constructive comments. We are glad to see the positive assessment of our paper and appreciate the detailed feedback.
>
>
> **Q1: Generation of the positional encoding $P_t$ given $t$ & Generalization to a longer time horizon than training.**
>
> A: SlotFormer performs rollout in an autoregressive manner. Given slots from the past $T$ steps $\\{S_t\\}\_{t=1}^T$, it adds sinusoidal positional encoding to them and leverages the Transformer to predict the next slots $\\hat{S}\_{T+1}$. Then, we concatenate the slots from the last $T-1$ steps $\\{S_t\\}\_{t=2}^T$ and the predicted $\\hat{S}\_{T+1}$ to predict $\\hat{S}\_{T+2}$, and repeat this process for further steps.
>
> As you can see, the input time window for the Transformer is always $T$, so we always need the same sinusoidal positional encoding of length $T$. Thanks to the autoregressive generation strategy, SlotFormer can predict longer time horizons than seen in training, without the positional encodings facing the out-of-distribution scenario.
>
>
> **Q2: Are the training video clips generated by random crops or if the full videos are used?**
>
> A: During training, we randomly crop short video clips with a fixed length $T + K$ from the full videos on the fly in data loaders. The full episodes are only used for evaluation. We have clarified this point in Section 4.2 of our revised paper.
>
>
> **Q3: Showing some generated videos from VQFormer to better compare with SlotFormer.**
>
> A: We have added the videos generated from VQFormer rollouts on the OBJ3D dataset in Figure 3, Figure 7, and Figure 12. Qualitatively, because of the strong VQ-VAE decoder, VQFormer is able to generate sharp videos without blurry objects. However, since it does not explicitly model each object, the simulated objects have inconsistent properties (e.g. color) and wrong dynamics. This proves that a strong decoder is not enough for learning multi-object dynamics.
>
>
> **Q4: Showing generated videos from SlotFormer on all the datasets.**
>
> A: We have uploaded the generated videos from SlotFormer to our [project page](https://slotformer.github.io/). This includes all four datasets we experiment on, namely, OBJ3D, CLEVRER, Physion, and PHYRE.
>
> Note that, as discussed in the paper, the generated videos on Physion are of low quality. This is because STEVE’s Transformer-based slot decoder is not designed for pixel space reconstruction. In the qualitative results, even STEVE's reconstructed videos (which have full access to the entire input videos) are visually unappealing. Nevertheless, SlotFormer is still able to simulate the motion of objects, such as collisions and falling, thus benefiting the VQA task.
>
>
> **Q5: Compare downstream tasks performance when using only burn-in frames and when doing rollout with SlotFormer.**
>
> A: Here we compare the performance of VQA and action planning tasks when the model only receives burn-in frames.
>
> VQA:
> - For CLEVRER predictive questions, as shown in Table 3, Aloe with only burn-in frames achieves 93.1% and 87.3% accuracy in the per-option and per-question settings. With explicit rollout, Aloe equipped with SlotFormer achieves 96.5% and 93.3% accuracy in the two settings, respectively;
> - For Physion, as shown in Table 4, the VQA accuracy with only burn-in frames is 65.2%, while with explicit rollout, the accuracy is 67.1%.
>
> Action Planning:
> - On PHYRE dataset, the AUCCESS when using only burn-in frames is 80.7, while with explicit rollout, the AUCCESS is 82.0. We have added the burn-in-only result to Table 5 and Section 4.4 of the revised paper as a baseline named *SAVi*.
>
>
> **Q6: Showing line plots for the ablation study with different burn-in and rollout length.**
>
> A: We thank the reviewer for the suggestion. We have created line plots for the ablation study regarding burn-in and rollout length in Figure 11. We also refer the readers to this result in Section 4.5 and Appendix E.5 of the revised paper.
>
>
> **Q7: Action-conditioned generation using SlotFormer.**
>
> A: Though we did not test action-conditioned generation with SlotFormer in our experiments, it can be easily adapted to this task. In [1], PARTS inputs the action token with slots to the predictor to perform action-conditioned generation. Similarly, when predicting future slots $\\hat{S}\_{T+1}$ conditioned on action $a_T$, we can also input $T$ past slots $\\{S_t\\}\_{t=1}^T$ and the action token $a\_T$ to our Transformer-based dynamics module. This enables spatio-temporal interactions between objects and the actions. We have added action-conditioned generation as a future direction of SlotFormer in the paper (see Appendix F of the revised paper).
>
> [1] Zoran, Daniel, et al. "PARTS: Unsupervised segmentation with slots, attention and independence maximization." ICCV. 2021.

---

> > ### Comment · Reviewer_yrhY · 2022-11-16
> > **Reply and Score Update**
> >
> > I thank the authors for their thorough responses! All my concerns were resolved satisfactorily. I update the score to 8 and recommend acceptance.

---

> > > ### Author Response · Authors · 2022-11-16
> > > **Re: Reply and Score Update**
> > >
> > > Thank you for updating your review! We are happy to hear that we were able to address all of your concerns.

---

### Official Review · Reviewer_eCuX · 2022-10-25

**Confidence:** 5
**Correctness:** 4
**Technical Novelty And Significance:** 3
**Empirical Novelty And Significance:** 3
**Recommendation:** 8

**Clarity, Quality, Novelty And Reproducibility:**

- I believe the combination of the two ideas of 1) extracting aligned slots from video with a pre-trained object discovery module and 2) treating the problem as a sequence modeling task produces a novel approach for the task of object-centric video prediction
- I think the reproducibility of this work overall is relatively low because of the amount of compute required (4 GPUs in less than 5 days). I would encourage the authors to release trained model weights when open-sourcing their code. However, the authors make a good effort to provide extensive experiment details in the appendix and the overall method is fairly simple + builds on top of existing methods (e.g., SAVi).
- Overall the paper is very well written. Moving the limitations section from the appendix to the main text is important, as well as expanding it to discuss, e.g., handling occlusion, + object appearance/disappearance within the burn-in frames, and deterministic vs. stochastic future prediction.
- Adding a description for SAVi, STEVE, and Aloe in the main text would help the readability of the paper. Also, I was missing the definition of the AUCCESS metric in the main text.

**Strength And Weaknesses:**

Strengths
=====
- The paper makes progress towards answering the difficult question of how to design an effective unsupervised approach for long-term video prediction capable of dealing with complex object-centric dynamics.
- The dynamics model has an elegant and simple design as a basic autoregressive transformer architecture. It appears to be simple enough that it is likely to be useful as a starting point for future methods that tackle more advanced tasks. For example, the dynamics module is shown to be capable of integrating with both SAVi and STEVE.
- The experiments are thorough; the paper uses proper baselines and metrics, multiple relevant multi-object video environments, and validates the design choices with strong empirical results.
- The paper provides a “bonus” insight I found particularly intriguing --- that a strong decoder (VQ-VAE) is not sufficient for learning complex multi-object dynamics.

Weaknesses
=====
I found just a few minor weaknesses to point out:
- It was a bit odd that OCVT was discussed as the most relevant closest work, yet results for this baseline were not provided. If I understand correctly from the appendix related work section, it consistently underperformed G-SWM on all benchmarks? Even so, it would be good to include results for OCVT.
- The simplicity of the approach hinges on the fact that all objects are visible at time step 1 and no new objects appear. This is a fairly important topic to discuss in the paper as this is limiting in terms of applying this approach to real-world video. What would it take to support handling occlusion and appearing/disappearing objects within the burn-in frames?
- Another weakness of the approach is that it is not trained end-to-end with the object discovery module which makes the training process cumbersome, and does not allow the object discovery to take advantage of long-term motion cues.


**Summary Of The Paper:**

This paper introduces an approach for learning to perform high-quality long-term video prediction via an object-centric latent bottleneck. The approach hinges on two key ideas. The first is to use a pre-trained temporal object discovery method to extract a sequence of aligned slots. The second is to use an autoregressive Transformer to rollout future slots conditioned on the sequence of aligned past slots. This enables the model to combine spatial and (multi-step) temporal information to improve the long-term accuracy of future slot rollouts. Extensive experiments exploring design choices, video prediction quality, and object-centric reasoning capabilities validate the efficacy of the approach.

**Summary Of The Review:**

Overall, I believe this paper contributes valuable insights as well as a useful method for the considered problem. I believe it is ready for publication in its current state.

---
Update after rebuttal: Maintaining my initial positive stance on the paper and recommend acceptance.

---

> ### Author Response · Authors · 2022-11-11
> **Response to Reviewer eCuX**
>
> We thank the reviewer for the detailed feedback and encouraging comments.
>
> **Q1: Does OCVT underperform G-SWM on all benchmarks? Can you include the results for the OCVT baseline.**
>
> A: Indeed, for the long-term video prediction task, OCVT underperforms G-SWM in 7 out of 8 cases (see OCVT [1] page 7, Figure 2 and Figure 3). Since SlotFormer outperforms G-SWM in all video prediction metrics, we did not include the result of OCVT in our initial submission.
>
> Since the code of OCVT is not released, we tried our best to reproduce it and train on our video prediction datasets during the rebuttal period. Due to the limited space, we include its results in Appendix E.3 of the revised paper, and refer the readers to it in Section 4.2. As expected, OCVT underperforms SlotFormer in all the metrics.
>
> [1] Wu, Yi-Fu, Jaesik Yoon, and Sungjin Ahn. "Generative Video Transformer: Can Objects be the Words?." ICML. 2021.
>
>
> **Q2: How can SlotFormer handle occlusions and appearing/disappearing objects within the burn-in frames?**
>
> A: We thank the reviewer for pointing out the confusion. We would like to highlight that it is not necessary for all objects to be visible at timestep 1, and SlotFormer can handle cases with new objects appearing during burn-in frames. [Here](https://i.imgur.com/bRxsF1F.png) we show three such examples on CLEVRER video prediction task, where a new object enters the scene during burn-in steps. Notably, SlotFormer is still able to simulate future frames with high quality and accurate object dynamics. We have added these examples and clarification texts to Appendix E.6 of the revised paper.
>
> SlotFormer is also able to handle occlusions/disappearing thanks to the spatio-temporal attention performed by the Transformer. Given $T$ burn-in frames, if an object is not visible at some steps, the Transformer can attend to frames where it is visible to infer its properties and motion. For example, in Figure 10 (left) of Appendix E.4 ([this figure](https://i.imgur.com/5k6PGe7.png)), the yellow sphere is severely occluded by the purple cube in $x_5, x_6$. As shown in the 4th row, SlotFormer attends to the yellow sphere in previous frames to infer its future states. We have highlighted this point in the revised paper.
>
>
> **Q3: End-to-end training of SlotFormer with the object discovery module.**
>
> A: This is indeed a limitation of our work. However, we would like to note that, in our preliminary experiments, joint training of the object-centric module and the dynamics module usually leads to training crashes. Also, pre-training SAVi/STEVE and training SlotFormer over slots both take lots of resources and time. Separating them makes hyper-parameter tuning easier and thus better reproducibility. But we agree that joint training is an interesting direction, which has been added to the future direction section (see Appendix F of the revised paper).
>
>
> **Q4: The reproducibility of SlotFormer & Releasing trained model weights with code.**
>
> A: As suggested by the reviewer, we will release our code as well as the trained model weights alongside the camera-ready version of this paper. We have added this to the Reproducibility Statement of the revised paper.
>
>
> **Q5: Moving the Limitation section to the main text & Expanding discussions.**
>
> A: We have expanded the Limitations section, and additionally discuss deterministic vs. stochastic future prediction. However, due to the limited space, we cannot move it to the main text, and instead point the readers to it in the Conclusion section (see Section 5 of the revised paper).
>
>
> **Q6: Adding descriptions for SAVi, STEVE, Aloe and the AUCCESS metric.**
>
> A: Thanks for the great suggestion. We have added descriptions for SAVi and STEVE in Section 4.1, Aloe in Section 4.3, and AUCCESS in Section 4.4 of the revised main text. We also refer the readers to Appendix C for complete details.

---

> > ### Comment · Reviewer_eCuX · 2022-11-18
> > **Thanks**
> >
> > Thanks for the response to my questions and concerns, particularly about the burn-in frames. I maintain my initial positive score.

---

> > > ### Author Response · Authors · 2022-11-18
> > > **Re: Thanks**
> > >
> > > We also want to thank the reviewer again for pointing out this confusion and for all the other insightful comments. They really help us improve our work!

---

### Author Response · Authors · 2022-11-11
**General Response**

We would like to thank the reviewers for their helpful feedback and insightful comments.

We are glad that the reviewers find our paper “*well written*” (eCuX), “*clear*” (yrhY), and “*easy to follow*” (HGtg), our dynamics modeling approach “*elegant*” (eCuX), “*effective*” (yrhY), and “*neat*” (HGtg); also, our experimental results are considered “*thorough*”, and show “*strong empirical results*” (eCuX), “*good future predictions*”, “*benefits on downstream tasks*” (yrhY), and “*the effectiveness of the method*” (HGtg). Furthermore, Reviewers eCuX and yrhY suggest that our method can become the “*starting point*” (eCuX)/“*go-to model*” (yrhY) for future research in this direction.

From the evaluation standpoint, the reviewers find the comparison to the VQFormer baseline insightful, i.e. it suggests that “*object-centric representation can be key for good learning of dynamics*” (yrhY), and “*a strong decoder (VQ-VAE) is not sufficient for learning complex multi-object dynamics*” (eCuX). Finally, reviewer eCuX confirms that our experiment “*uses proper baselines and metrics, multiple relevant multi-object video environments*”, and reviewer HGtg also agrees that our paper “*is novel in the configuration of the experiments*”.

We further thank the reviewers for their constructive feedback. In response, we have conducted two new baseline comparisons which we summarize below:

1. **Include the results for OCVT [1] (suggested by reviewer eCuX)**

Since the OCVT code is not released, we reproduce the model, and train it on OBJ3D and CLEVRER in the video prediction task. Due to the limited space, we include the results in Appendix E.3 of the revised paper. As expected, OCVT underperforms SlotFormer in all the metrics. This is because OCVT builds upon object slots from SPACE [2], which applies heavy priors and fails to work on visually complex scenes. In addition, OCVT requires explicit Hungarian matching for slot alignment in loss computation, while SlotFormer naturally enjoys the temporal alignment of slots from SAVi.


2. **Add a baseline with RNN-based model over slots (suggested by reviewer HGtg)**

To better convey the intuition of why our method excels in long-term modeling, we design a baseline that runs the Transformer-LSTM dynamics module over the same pre-trained slots as SlotFormer. It adopts the same Transformer as SlotFormer, but only takes in slots at a single timestep, thus only modeling the immediate spatial interaction of objects. This is followed by a per-slot LSTM to learn the temporal dynamics. We test this model in the video prediction task on OBJ3D and CLEVRER datasets. The results are shown in Appendix E.3. SlotFormer outperforms this baseline in all the metrics, proving that our Transformer that performs joint spatio-temporal reasoning over a large context window is the key to SlotFormer’s success.


[1] Wu, Yi-Fu, Jaesik Yoon, and Sungjin Ahn. "Generative Video Transformer: Can Objects be the Words?." ICML. 2021.

[2] Lin, Zhixuan, et al. "SPACE: Unsupervised Object-Oriented Scene Representation via Spatial Attention and Decomposition." ICLR. 2019.


We have uploaded a revised submission which includes two additional experiments and various clarifications to address the feedback from the reviewers. For ease of review, we highlight the revised text in blue. For other questions raised by the reviewers, please see our response to individual questions and concerns below each review.

---

### Author Response · Authors · 2022-11-15
**Author/reviewer Discussions**

Dear Reviewers,

Thank you for providing insightful feedback to help us improve our work. We have addressed all the questions in detail and revised the manuscript. Please have a look at our response to each question and the revised manuscript. Please let us know if there is anything that needs further clarifications. Thank you.

Best Regards,
Authors

---

### Decision · Program_Chairs · 2023-01-20

**Decision:**

Accept: poster

**Justification For Why Not Higher Score:**

The main idea of the proposed model is not very novel.

**Justification For Why Not Lower Score:**

The paper is very well written, the proposed model is simple (in a good sense) and works well. The experiments are very comprehensive.

**Metareview: Summary, Strengths And Weaknesses:**

The authors propose a Transformer-based, object-centric autoregressive model for consistent future prediction. The method first learn to extract object-centric representations via pretraining. By using SAVi/STEVE, these slots extracted across multiple frames contain alignment information. Then, it trains a Transformer-based autoregressive model that takes the object slots as input. The experiment is comprehensive and demonstrate good performance compared to the previous methods.

Strengths. First of all, the paper is very clear and well written. The experiment is very comprehensive. The experiment results are also strong. The idea of the proposed method is not very novel but the simple-and-well-working can be considered also a strong merit that can cover the weakness of the idea novelty.

The main weakness is the novelty of the idea. But as I mentioned, I believe the paper still contains other important contribution to the community. The model may be trained end-to-end but I agree that it can be a future work or not that easy.

**Note From Pc:**

if the above contains the word "oral" or "spotlight" please see: "oral" presentation means -> notable-top-5% and "spotlight" means -> notable-top-25%. As stated in our emails, we are disassociating presentation type from AC recommendations